

# A robust error correction method for numerical weather
# prediction wind speed based on Bayesian optimization,
# Variational Mode Decomposition, Principal Component
# Analysis, and Random Forest: VMD-PCA-RF (version
# 1.0.0)
Shaohui Zhou[1], Chloe Yuchao Gao[2*], Zexia Duan[1], Xingya Xi[3], and Yubin Li[1]
[1]Collaborative Innovation Centre on Forecast and Evaluation of Meteorological Disasters, Key
Laboratory for Aerosol-Cloud-Precipitation of China Meteorological Administration, School of
Atmospheric Physics, Nanjing University of Information Science and Technology, Nanjing, 210044,
China.
[2]Department of Atmospheric and Oceanic Sciences and Institute of Atmospheric Sciences, Fudan
University, Shanghai, 200438, China.
[3]School of Atmospheric Sciences, Sun Yat-sen University, and Southern Marine Science and
Engineering Guangdong Laboratory (Zhuhai), Zhuhai, 519082, China
*Correspondence to*: gyc@fudan.edu.cn
**Abstract.** Accurate wind speed prediction is crucial for the safe utilization of wind resources. However,
current single-value deterministic numerical weather prediction methods employed by wind farms do
not adequately meet the actual needs of power grid dispatching. In this study, we propose a new hybrid
forecasting method for correcting 10-meter wind speed predictions made by the Weather Research and
Forecasting (WRF) model. Our approach incorporates Variational Mode Decomposition (VMD),
Principal Component Analysis (PCA), and five artificial intelligence algorithms: Deep Belief Network
(DBN), Multilayer Perceptron (MLP), Random Forest (RF), eXtreme Gradient Boosting (XGBoost),
light Gradient Boosting Machine (lightGBM), and the Bayesian Optimization Algorithm (BOA). We
first construct WRF-predicted wind speeds using the Global Prediction System (GFS) model output
based on prediction results. We then perform two sets of experiments with different input factors and
apply BOA optimization to debug the four artificial intelligence models, ultimately building the final
models. Furthermore, we compare the forementioned five optimal artificial intelligence models suitable
for five provinces in southern China in the wintertime: VMD-PCA-RF in December 2021 and
VMD-PCA-lightGBM in January 2022. We find that the VMD-PCA-RF evaluation indexes exhibit
relative stability over nearly a year: correlation coefficient (R) is above 0.6, accuracy rate (FA) is above
85 %, mean absolute error (MAE) is below 0.6 m/s, root mean square error (RMSE) is below 0.8 m/s,



relative mean absolute error (rMAE) is below 60 %, and relative root mean square error (rRMSE) is
below 75 %. Thus, for its promising performance and excellent year-round robustness, we recommend
adopting the proposed VMD-PCA-RF method for improved wind speed prediction in models.
**1 Introduction**
Sustainable energy plays a vital role in reducing carbon footprint and increasing system reliability
(Hanifi et al., 2020). As renewable energy sources have a negligible carbon footprint, they have
become the preferred choice for many industries in the power sector (Dhiman and Deb, 2020). Among
these sources, wind energy is a crucial low-carbon energy technology with the potential to become a
sustainable energy source (Tascikaraoglu and Uzunoglu, 2014). In 2022, the global wind power
capacity reached 906 GW, with a 9 % year-on-year increase due to a newly installed capacity of 77.6
GW. The global onshore wind market increased by 68.8 GW, while facing a 5 % growth decline
compared to the previous year. Such change is attributed to a slowdown in China and the U.S., the
world's two largest wind markets that account for over two-thirds of the world's onshore wind farm
installations (Joyce and Feng, 2023). Therefore, accurate and stable wind speed prediction (WSP) is
very important for the safe and stable operation of the power grid system and improving the utilization
rate of wind energy and economic development (Guo et al., 2021; Xiong et al., 2022; Tang et al.,

48  2021).

Current WSP algorithms are primarily categorized into physical algorithms (Zhao et al., 2016),
statistical algorithms (Wang and Hu, 2015; Barthelmie et al., 1993), machine learning (ML) algorithms
(Huang et al., 2019; Salcedo-Sanz et al., 2011; Ma et al., 2020), and hybrid algorithms (Deng et al.,
2020; Xu et al., 2021; Zhao et al., 2019; Xiong et al., 2022; Tang et al., 2021). Physical methods, such
as numerical weather prediction (NWP), are commonly used in wind speed forecasting. NWP, which
accounts for atmospheric processes and physical laws, solves discrete mass, momentum, and energy
conservation equations along with other fundamental physical principles, establishing itself as a widely
adopted and reliable physical method. Currently, the High-resolution Limited Area Model (HIRLAM)
(Landberg, 1999), the European Center for Medium-Range Weather Forecast (ECMWF) model, the
fifth-generation mesoscale model (MM5) (Salcedo-Sanz et al., 2009), and the Weather Research and
Forecasting Model (WRF) (Prósper et al., 2019) are extensively utilized for wind speed prediction.



However, NWP modeling faces challenges due to the selection of parameterization schemes, such as
model microphysics and systematic errors, which exhibit temporal and spatial differences and
uncertainties. These uncertainties hinder the accuracy of NWP models in wind speed prediction,
making it difficult to meet the rising demands of the grid system (Zhao et al., 2019; Xu et al., 2021).
Studies have demonstrated that enhancing the accuracy of numerical weather prediction (NWP)
models and correcting prediction errors can effectively minimize the errors associated with wind speed
prediction. These research endeavors have typically sought to optimize the physical and dynamic
parameters of the NWP model (Cheng et al., 2013), refine the model structure (Jiménez and Dudhia,
2012), or improve the accuracy of model inputs through preprocessing and denoising techniques (Xu et
al., 2015). Additionally, improving initial field error through methods, such as target observation and
data assimilation (Williams et al., 2013), can also minimize wind speed errors predicted by NWP
models.
Physical methods are generally more appropriate for long-term wind speed prediction, such as
those 48-72 hours in advance, while their practical application in short-term forecasting is limited
(Zhao et al., 2019; Deng et al., 2020; James et al., 2018). In contrast, statistical methods utilize
historical data to establish a relationship between input and output variables and are therefore
well-suited for short-term wind speed prediction. They are usually time series models, such as
Autoregressive Moving Average (ARMA) (Erdem and Shi, 2011) and Autoregressive Integrated
Moving Average (ARIMA) (Wang and Hu, 2015). Whereas filtering models (Cassola and Burlando,
2012; Chen and Yu, 2014), machine learning models (Hu et al., 2013), and hybrid models (Huang et al.,
2019) have been gradually developed to further improve wind speed prediction accuracy.
With purely statistical models becoming less suitable for wind speed predictions beyond 6 hours,
the use of a combination of physical and statistical methods has gained growing interest (Zjavka, 2015;
Xu et al., 2021). The error correction model improves the accuracy of the NWP model by training the
relationship between the NWP predictor variables and the observed correlation variables (Sun et al.,
2019). However, traditional error prediction models rely solely on historical wind speed sequences as
input factors (Deng et al., 2020; Guo et al., 2021) and do not incorporate the characteristic
meteorological factors forecasted by the WRF model. Studies have shown that considering all relevant
historical meteorological factors can lead to more accurate predictions compared to only taking into



89 account historical wind speed (Zhang et al., 2019c). Therefore, it is crucial to include meteorological

90 characteristic factors as input in the prediction model.

91  For an error prediction model, wind speed is the most important input factor. Traditionally, the

92 error prediction model uses historical wind speed data as input, without any feature selection. Feature

93 selection methods, such as filtering methods, are commonly used in time series analysis. Currently,

94 empirical mode decomposition (EMD) (Liu et al., 2018; Guo et al., 2012), ensemble empirical mode

95 decomposition (EEMD) (Wang et al., 2017), wavelet decomposition (WD) (Zhang et al., 2019b),

96 variational mode decomposition (VMD) (Hu et al., 2021; Zhang et al., 2019a), and other filtering

97 methods are used to select key features in the wind speed data. As mentioned above, studies have

98 shown that these feature selection methods can effectively extract the hidden features in the wind speed

99 series to improve wind speed prediction accuracy. However, despite the effectiveness of wind speed

100 filtering methods in wind speed prediction, only a few studies have applied these methods to the

101 correction of wind speed errors in NWP forecasting (Xu et al., 2021; Li et al., 2022).

102  In addition, traditional error correction methods generally adopt linear regression (Dong et al.,

103 2013), multiple linear regression (Liu et al., 2016), machine learning (Salcedo-Sanz et al., 2011), and

104 deep learning algorithms (Zhang et al., 2019c). However, the efficacy of machine learning and deep

105 learning algorithms is highly dependent on the selection of model parameters (Guo et al., 2021; Xiong

106 et al., 2022). The Bayesian optimization algorithm (Li and Shi, 2010; Guo et al., 2021) is considered a

107 relatively advanced algorithm for optimizing model parameters and has been widely used in MATLAB

108 and Python packages.

109  In this study, we investigate a multi-step wind speed forecasting model that combines NWP

110 simulation and an error correction strategy. We present two sets of experiments divided into three steps:

111 (1) we use the first group of experiments to extract hidden features from various meteorological

112 elements forecasted by NWP; The second group of experiments mainly focuses on the wind speed

113 forecast of NWP, and the VMD-PCA algorithm is used to extract the hidden features in the forecasted

114 wind speed; each set of experimental input factors is matched with the actual 10-meter wind speed data

115 of 410 stations in time and space; (2) we employ four advanced machine learning algorithms optimized

116 by the BOA algorithm, and DBN deep learning algorithm to train the two groups of experiments and

117 perform 5-fold cross-validation; and (3) we analyze six distinct wind speed error indicators to compare



and identify the most suitable wind speed error correction schemes for the five southern provinces in
winter and throughout most of the year. The remainder of this paper is organized into sections
discussing the effects of the BOA-VMD-PCA approach, the interpretability of RF feature importance,
and the stability analysis of the proposed models.
**2 Data and methods**
**2.1 Data**
The target observation data includes 2-m air temperature, 2-m specific humidity, 10-meter wind
speed, surface pressure, and precipitation. These data are collected on equivalent latitude and longitude
grid scale, primarily from five provinces in China: Guangdong, Guangxi, Yunnan, Guizhou, and
Hainan, covering a geographical range of 15-32.97°N and 94-120.97°E. The spatial resolution of the
grid is 0.03° × 0.03° and the temporal resolution is 1 hour. The dataset is constructed through the
integration of multiple sources, including ground and satellite data, and is refined using advanced
techniques such as multi-grid variational assimilation, physical inversion, and terrain correction. This
dataset exhibits superior quality in comparison to other products, offering higher spatial and temporal
resolutions. For the purposes of this paper, the 10-meter wind speed data is interpolated across 410
sites, as illustrated in Figure 1.

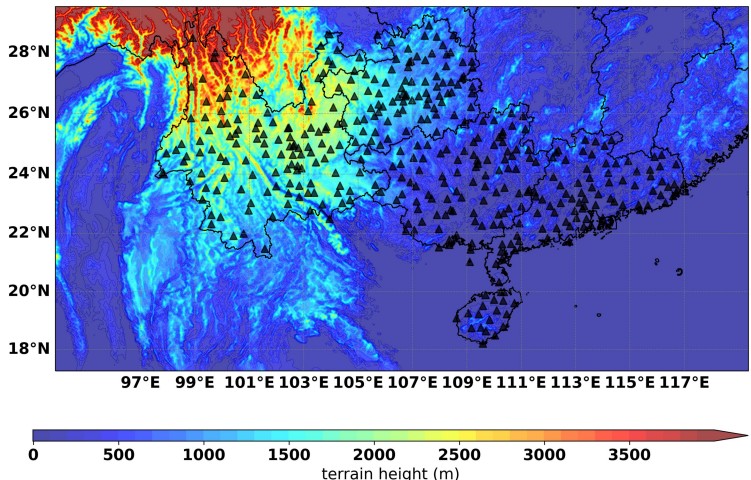


**Figure 1: The elevation map of the five southern provinces in china (black triangles represent weather**
**stations).**




## 2.2 Methods

### 2.2.1 WRF simulation

The WRF 4.2 model, developed by the United States' National Center for Environmental Prediction (NCEP), represents a new generation of mesoscale numerical models with numerous applications in research forecasting. The WRF model, written in Fortran 90 language, offers advantages such as portability, scalability, and high efficiency. It employs Arakawa C-grid points in the horizontal direction and terrain-following mass coordinates in the vertical direction. When forecasting meteorological elements, the WRF model uses the US Global Weather Forecast Data (GFS) developed by NCEP and the National Center for Atmospheric Research (NCAR). The GFS system includes data related to the atmosphere and land variables, such as temperature, precipitation, and wind data. The system is updated every 6 hours, at 0:00, 6:00, 12:00, and 18:00 UTC, and provides predictions for the subsequent eight days. Given that the time scale of the meteorological station data in the study area is 1 hour, the forecast data time interval of the WRF model is also set to 1 hour. As a widely used numerical weather forecast model, the WRF model is suitable for weather studies from a few meters to several thousand kilometers. Therefore, this paper uses the WRF model to predict 10-meter wind speed as the input factor for the error correction model (Xu et al., 2021).

Using the WRF model in combination with daily data resolution of 0.25° × 0.25°, the model initiates at 18:00 UTC and generates forecasts every 3 hours for a total duration of 102 hours. The regular Global Forecast System (GFS) forecast field data serve as the initial field and lateral boundary conditions for the WRF model. Surface static data, such as terrain, soil data, and vegetation coverage, are derived from the Moderate Resolution Imaging Spectroradiometer (MODIS) satellite with a resolution of 15 seconds (approximately 500 meters). Incorporating a two-layer grid nesting configuration, the forecast area is illustrated in Figure 2. The grid dimensions are 600×500 and 967×535, with horizontal grid resolutions of 9 km and 3 km, respectively. The grid center points are set at 29°N and 96°E. The "CONUS" parameterization scheme is used, including the Thompson microphysics scheme, the Tiedtke cumulus parameterization scheme, the RRTMG long-wave and short-wave radiation schemes, the Mellor-Yamada-Janjić (MYJ) boundary layer and near-surface parameterization schemes, and the MYJ surface layer parameterization scheme. The Noah Land



Surface Model (LSM) is utilized for the surface process plan, generating a WRFOUT numerical
weather forecast file including meteorological elements such as temperature, humidity, and
precipitation. The WRF configuration process is detailed in Table 1.

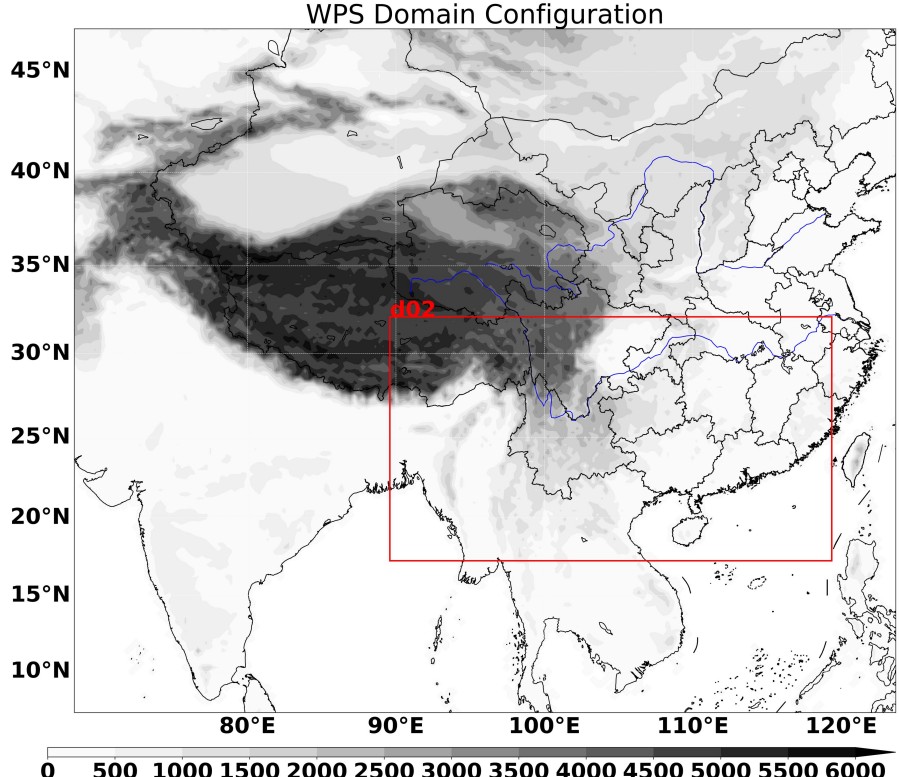


**Figure 2: Schematic diagram of the simulation area of the WRF model.**

**Table 1: WRF configuration scheme**

| Model (Version) | WRF (V4.2) | |
| --- | --- | --- |
| Domains | D1 | D2 |
| Horizontal grid points | 600*500 | 967*535 |
| Δx (km) | 9 | 3 |
| Vertical layers | 58 | |
| Longwave radiation | RRTMG (Iacono et al. 2008) | |
| Shortwave radiation | RRTMG (Iacono et al. 2008) | |
| Land surface | Noah LSM (Chen et al. 1997) | |
| Surface layer | MYJ (Janjic 1994) | |
| Microphysics | Thompson (Thompson et al. 2008) | |



| Boundary layer | MYJ (Janjic 1994) |
|---|---|
| Cumulus | Tiedtke (Tiedtke 1989, Zhang et al. 2011) |


**2.2.2 Variational mode decomposition**
As a new filtering method, VMD is robust in feature selection. The VMD algorithm decomposes a
time series signal into several intrinsic mode functions (Isham et al., 2018). The sum of the modes
equals the original signal, and the sum of the bandwidths is the smallest. The analysis signal is
calculated using the Hilbert transform to estimate the modal bandwidth. The optimization model is
described as
$$\left\{ \min_{\{u_k\},\{\omega_k\}} \left\{ \sum_{k=1}^{K} \| \partial_t \left[ \left( \delta(t) + \frac{j}{\pi t} \right) u_k(t) \right] e^{-j\omega_k t} \Big|_2^2 \right\} \ s.t. \ \sum_{k=1}^{K} u_k = v \right\} \qquad (1.1)$$

where $K$ is the total number of modes, $u_k$ is the decomposed $K$-th mode, $w_k$ is the corresponding
center frequency, and $v$ is the time-series signal, representing the wind speed sequence predicted by the
WRF model in this study.
The above constrained problem can be transformed into an unconstrained problem using the
Lagrangian function:
$$L\left(\{u_k\},\{\omega_k\},\lambda\right) = \omega_k^{n+1} = \frac{\int_0^\infty \omega |\hat{u}_k(\omega)|^2 d\omega}{\int_0^\infty |\hat{u}_k(\omega)|^2 d\omega} \sum_{k=1}^K \| \partial_t \left[ \left( \delta(t) + \frac{j}{\pi t} \right) u_k(t) \right.$$


$$\times ] e^{-j\omega_k t} \|_2^2 + \| v(t) - \sum_{k=1}^K u_k(t) \|_2^2 + \left\langle \lambda(t), v(t) - \sum_{k=1}^K u_k(t) \right\rangle \qquad (1.2)$$

where $\alpha$ is the penalty parameter and $\lambda(t)$ is the Lagrange multiplier.
Then we update $u_k$, $w_k$, and $\lambda$ using the alternating direction method of the multiplier:
$$\hat{u}_k^{n+1}(\omega) = \frac{\hat{v}(\omega) - \sum_{i\neq k} \hat{u}_i(\omega) + \frac{\hat{\lambda}(\omega)}{2}}{1 + 2\alpha \left( \omega - \omega_k \right)^2} \qquad (1.3)$$

$$\omega_k^{n+1} = \frac{\int_0^\infty \omega |\hat{u}_k(\omega)|^2 d\omega}{\int_0^\infty |\hat{u}_k(\omega)|^2 d\omega} \qquad (1.4)$$

$$\hat{\lambda}^{n+1}(\omega) = \hat{\lambda}^n(\omega) + \tau \left[ \hat{v}(\omega) - \sum_{k=1}^K \hat{u}_k^{n+1}(\omega) \right] \qquad (1.5)$$



where $\tau$ is the update parameter.
When the accuracy (left side of the following expression) meets the following condition, $u_k$, $w_k$
and $\lambda$ would stop updating:
$$\sum_{k=1}^{K} \frac{\parallel \hat{u}_k^{n+1} - \hat{u}_k^n \parallel_2^2}{\parallel \hat{u}_k^n \parallel_2^2} < \varepsilon \qquad (1.6)$$

where $\varepsilon$ is the tolerance of the convergence criterion.
The VMD algorithm is implemented to decompose the wind speed signal predicted by the WRF
model. When using multiple sub-signals instead of the original signal, more features of the wind speed
can be obtained. Therefore, it is beneficial to improve the prediction accuracy when using the
sub-signal as input to the error correction model (Xu et al., 2021; Li et al., 2022).
**2.2.3 Principal Component Analysis**
Subsequences obtained by VMD usually have several illusory components. Using PCA to extract
the principal components of subsequences increases the number of features input to the model and
reduces the dimension of the data decomposed by VMD. When pcs are used as the input of the error
prediction algorithm, the pcs fully reflect the characteristics of the subsequence and reduce the model
complexity. The pcs $y_k$, $k=1, 2, …, K$ of the subsequence matrix U and the cumulative contribution rate
$\eta_n$ of first $n$ principal components are expressed as:
$$y_k = c_k' U \qquad (1.7)$$

$$\eta_n = \frac{\sum_{k=1}^{n} \lambda_k}{\sum_{k=1}^{K} \lambda_k} \qquad (1.8)$$

where $c_k$ is the corresponding characteristic unit vector, with $k=1, 2, …, K$; $\lambda_k$ is the characteristic
root, with $\lambda_1 \geq \lambda_2 \geq … \geq \lambda_K$.
**2.2.4 Proposed hybrid forecasting algorithms**
This study used five machine learning algorithms to conduct ten experiments across two main
paths. The first path involves increasing the variables related to wind speed in the forecast field, while
the second path entails extracting potential characteristic information of the forecast wind speed
through VMD and PCA and reducing the characteristic quantity of other forecast data. The overarching



goal is to achieve accurate correction of the forecast field wind speed. The flowchart of the artificial
intelligence models used to correct the WRF predicted wind speed for the two main experimental paths
is illustrated in Figure 3 and comprises the following three steps:

222        Step 1. Data fusion, cleaning, and standardization: As depicted in Figure 3, this paper proposes

two distinct experimental paths, with the primary difference being the selection of input variables. In
Experiment 1, as shown in Figure 6(c), 12 sets of data are selected from the WRF forecast field,
including altitude, 10-meter wind speed, latitude, longitude, surface pressure, relative humidity,
10-meter meridional wind, 10-meter zonal wind, 2-meter temperature, 2-meter dew point temperature,
10-meter wind direction, and hourly precipitation. Experiment 2, as illustrated in Figure 6(d), derives 8
sets of data by reducing the selected WRF field forecast data, including altitude, 10-meter wind speed,
latitude, longitude, surface pressure, relative humidity, 2-meter temperature, and hourly precipitation.
The focus is on unearthing hidden characteristic information of forecast wind speed. In this experiment,
the wind speed is decomposed into 9 Intrinsic Mode Functions (IMF) using VMD. Subsequently, a
low-dimensional wind speed vector is extracted from the 9 IMF components via PCA dimensionality
reduction, and all data are concatenated to construct the input factors for the model in Experiment 2.
Missing and outlier values are removed from the dataset. The two experiments standardize 12 sets of
meteorological elements (8 sets of meteorological elements in Figure 4, 9 IMF components, and three
PCA vectors in Figure 5) and wind speed observation data, respectively. Standardization addresses the
issue of varying meteorological factor values during training, which may result in different
contributions. In this paper, the 24-hour forecast data correspond to the observation data of the
subsequent 24 hours. The dataset spans from 00:00 on December 1, 2021, to 23:00 on February 28,
2022, totaling 2160 hours and encompassing 410 weather stations. Consequently, the original dataset
comprises 2160*410 samples, with each sample containing 12 meteorological features in Experiment 1
and 20 input features in Experiment 2.

243        Step 2. BOA optimization of AI models and cross-validation: In this study, the dataset is

partitioned into training, validation, and test sets in accordance with the time series. February 2022
serves as the training and validation sets, while December 2021 and January 2022 constitute the test set.
The training and validation sets are divided based on five-fold cross-validation. Both experiments
employ five machine learning algorithms (DBN, MLP, RF, XGBoost, and LightGBM) to construct



distinct machine learning models. Concurrently, this paper utilizes the BOA algorithm to tune the
parameters of all models, except for DBN, resulting in the optimal hyperparameters for each model.

250        Step 3. Model evaluation and error analysis: The trained machine learning models are applied to

the test set to obtain the revised wind speed data, and ultimately, the accuracy of all models is assessed
through the wind speed evaluation index. The ultimate goal here is to identify the best wind speed
correction model suitable for the entire year. Accordingly, the generalization of all models is evaluated
across other seasonal months of the year, culminating in the selection of the best model.

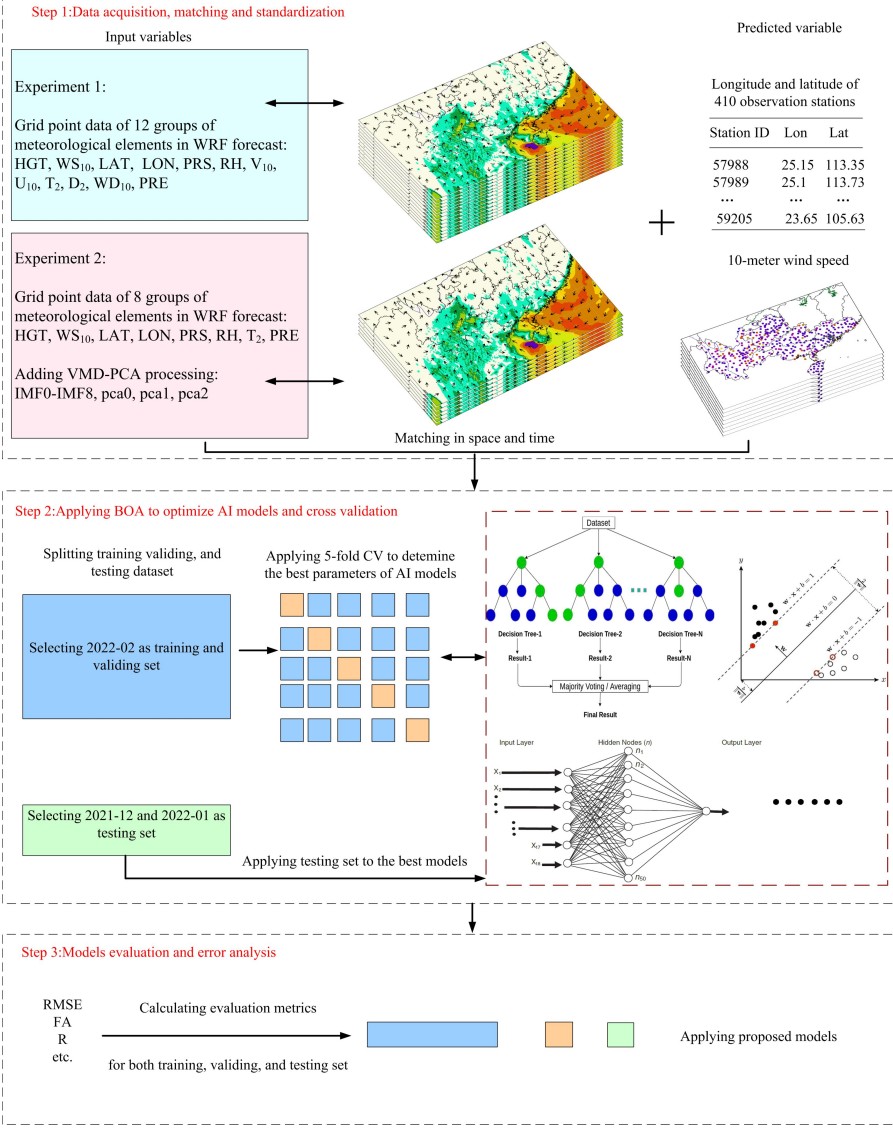





**Figure 3: Flowchart of the AI model used to correct WRF-predicted wind speeds in the two main experimental pathways.**


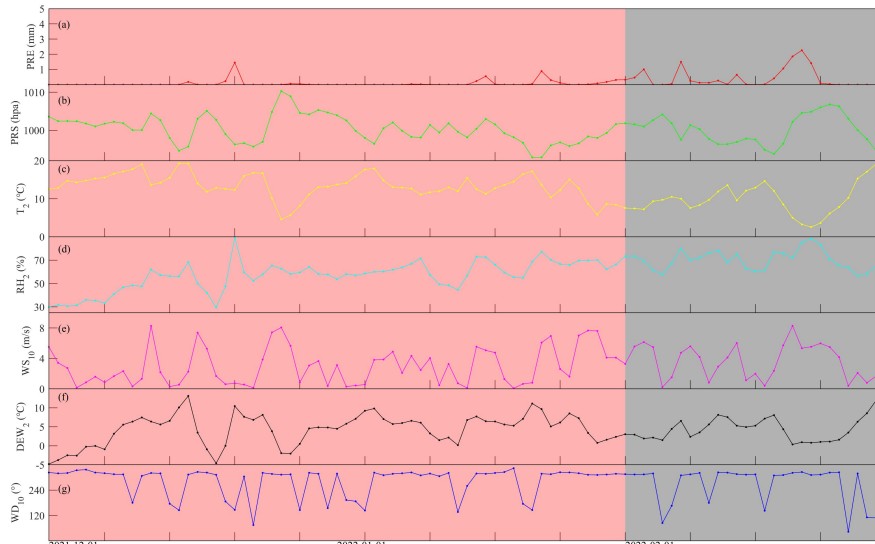


**Figure 4: Daily average hourly rainfall (a), surface pressure (b), 2-meter temperature (c), 2-meter relative humidity (d), 10-meter wind speed (e), 2-meter dew point temperature (f), and 10-meter wind direction ( g) which are located at Guangdong Lechang Station from December 1, 2021, to February 28, 2022. (February 2022 represents the training and verification sets, and December 2021 to January 2022 represents the testing set).**







**Figure 5: Three-dimensional view of 12 wind speed components after VMD and PCA processing of the**
**10-meter forecast wind speed at Lechang Station in Guangdong from December 1, 2021, to February 28,**
**2022.**

**2.2.5 Evaluation indicators**

273       There are many commonly used predictive effect evaluation indicators. This article uses the

following evaluation indicators: correlation coefficient (R), root mean square error (RMSE), mean
absolute error (MAE), relative root mean square error (rRMSE), relative mean absolute error (rMAE),
percentage of absolute error not greater than 1 m/s (FA). Six error indicators are used to evaluate the
correction results of short-term wind speed forecasts of wind farms. The formula for calculating the
error index is as follows:
$$R = \frac{\sum_{i}^{n}(y_i - \bar{y})(\hat{y}_i - \bar{\hat{y}})}{\sqrt{\sum_{i=1}^{n}(y_i - \bar{y})^2}\sqrt{\sum_{i=1}^{n}(\hat{y}_i - \bar{\hat{y}})^2}} \qquad (1.9)$$

$$RMSE = \sqrt{\frac{1}{n}\sum_{i=1}^{n}(\hat{y}_i - y_i)^2} \qquad (1.10)$$

$$MAE = \frac{1}{n}\sum_{i=1}^{n}|\hat{y}_i - y_i| \qquad (1.11)$$



$$rRMSE = \left[ \sqrt{\frac{1}{n}\sum_{i=1}^{n}(\hat{y}_i - y_i)^2} \Big/ \left( \frac{1}{n}\sum_{i=1}^{n} y_i \right) \right] \times 100\%$$ (1.12)

$$rMAE = \left( \frac{1}{n}\sum_{i=1}^{n}|\hat{y}_i - y_i| \Big/ \left( \frac{1}{n}\sum_{i=1}^{n} y_i \right) \right) \times 100\%$$ (1.13)

$$FA = N_r / N_f$$ (1.14)

Among them, $n$ represents the number of samples, $\hat{y}_i$ represents the $i$-th predicted value, $y_i$
represents the $i$-th actual value; $N_r$ represents the number of wind speed absolute errors not greater than
1 m/s, and $N_f$ represents the number of research samples.

**3 Results**
**3.1 Experiment 1 evaluation**
In Experiment 1, the BOA optimization algorithm was applied to five AI models to correct the
10-meter wind speed forecasted by WRF. There were 12 meteorological element features to establish
five different AI models (see Table 2 for the hyper-parameters of the five AI models). The training,
validation, and testing results for 10-meter wind speed are shown in Figures S1-5 in the supplementary
material. The RMSE values between the predicted and the observed value of the training set (validation
set) in the lightGBM, XGBoost, RF, DBN, and MLP models are 0.41 m/s (0.54 m/s), 0.31 m/s (0.56
m/s), 0.52 m/s (0.57 m/s), 0.59 m/s (0.62 m/s) and 0.73 m/s (0.73 m/s). The FA are 0.98 (0.94), 0.99
(0.93), 0.94 (0.93), 0.92 (0.91), and 0.88 (0.88). The R squared are 0.87 (0.77), 0.92 (0.75), 0.79 (0.73),
0.72 (0.69), and 0.57 (0.57). It is evident that all models, except the DBN model, can fit the training set
data well. The DBN model exhibits the weakest performance on both the training and validation sets.
Alternatively, the LightGBM and XGBoost models demonstrate superior prediction performance on
the training set compared to the validation set. The scatter points of the training sets of these two
models accumulate on the 1:1 diagonal, indicating slight overfitting. The RMSE of lightGBM,
XGBoost, RF, DBN, and MLP models on the test set in December 2021 (January 2022) are 0.67 m/s
(0.64 m/s), 0.70 m/s (0.67 m/s), 0.65 m/s (0.64 m/s), 0.77 m/s (0.74 m/s), and 0.74 m/s (0.68 m/s)
respectively. The FA of models on the test set in December 2021 (January 2022) are 89.68 %
(91.11 %), 87.90 % (89.88 %), 90.64 % (91.36 %), 86.74 % (87.71 %), and 86.08 % (89.57 %). The R



are 0.79 (0.77), 0.77 (0.75), 0.81 (0.78), 0.71 (0.68), and 0.75 (0.74). Considering different evaluation
indexes, the revision effects of the five models in two months demonstrate that RMSE is that January
2022 is generally lower than December 2021; FA is that January 2022 is generally higher than
December 2021; R is that January 2022 is generally lower than December 2021. Overall, the prediction
performance of the five models in January 2022 surpassed that in December 2021. Furthermore, the
LightGBM and RF models exhibited the best performance among the five models in the two-month test
sets, while the DBN model had the least effective correction effect.
With respect to the importance of RF characteristics (Fig.6a, c), it is indisputable that the 10 m
wind speed predicted by WRF plays a dominant role in correcting the actual wind speed. The ones
following are latitude, longitude and topographic height, which represent spatial geographic
information, and the actual wind speed is closely related to geographic information. Subsequently,
relative humidity is of lesser importance. The distribution of the humidity field typically correlates with
the movement of the atmosphere, which is also closely related to wind speed. Certain meteorological
elements, such as rainfall, 2 m dew-point temperature, and 2 m temperature, contribute less importance.

**Table 2. The best hyper-parameters of the models**

| Model | parameters |
|---|---|
| VMD-PCA-lightGBM | 'max_depth' : 28, 'min_child_samples' : 30, 'n_estimators' : 436, 'num_leaves' : 287 |
| VMD-PCA-XGBoost | 'gamma' : 1, 'max_depth' : 19, 'min_child_weight' : 1, 'n_estimators': 408 |
| VMD-PCA-RF | 'max_depth' : 31, 'max_features' : 14, 'min_samples_leaf' : 28, 'min_samples_split' : 3, 'n_estimators' : 371 |
| VMD-PCA-DBN | 'input_length' : 20, 'output_length' : 1, 'loss_function' : 'MSE', 'optimizer' : 'Adam', 'hidden_units' : [400, 200], 'batch_size' :20000, 'epoch_pretrain' : 100, 'epoch_finetune' : 200 |
| VMD-PCA-MLP | 'batch_size' : 10114, 'hidden_layer_sizes' : 305, 'max_iter' : 386 |
| lightGBM | 'max_depth' : 21, 'min_child_samples' : 19, 'n_estimators' : 312, 'num_leaves' : 297 |



| XGBoost | 'gamma' : 0, 'max_depth' : 21, 'min_child_weight' : 9, 'n_estimators': 299 |
| RF | 'max_depth' : 40, 'max_features' : 12, 'min_samples_leaf' : 23, 'min_samples_split' : 2, 'n_estimators' : 440 |
| DBN | 'input_length' : 12, 'output_length' : 1, 'loss_function' : 'MSE', 'optimizer' : 'Adam', 'hidden_units' : [400, 200], 'batch_size' : 20000, 'epoch_pretrain' : 100, 'epoch_finetune' : 200 |
| MLP | 'batch_size' : 10232, 'hidden_layer_sizes' : 494, 'max_iter' : 311 |


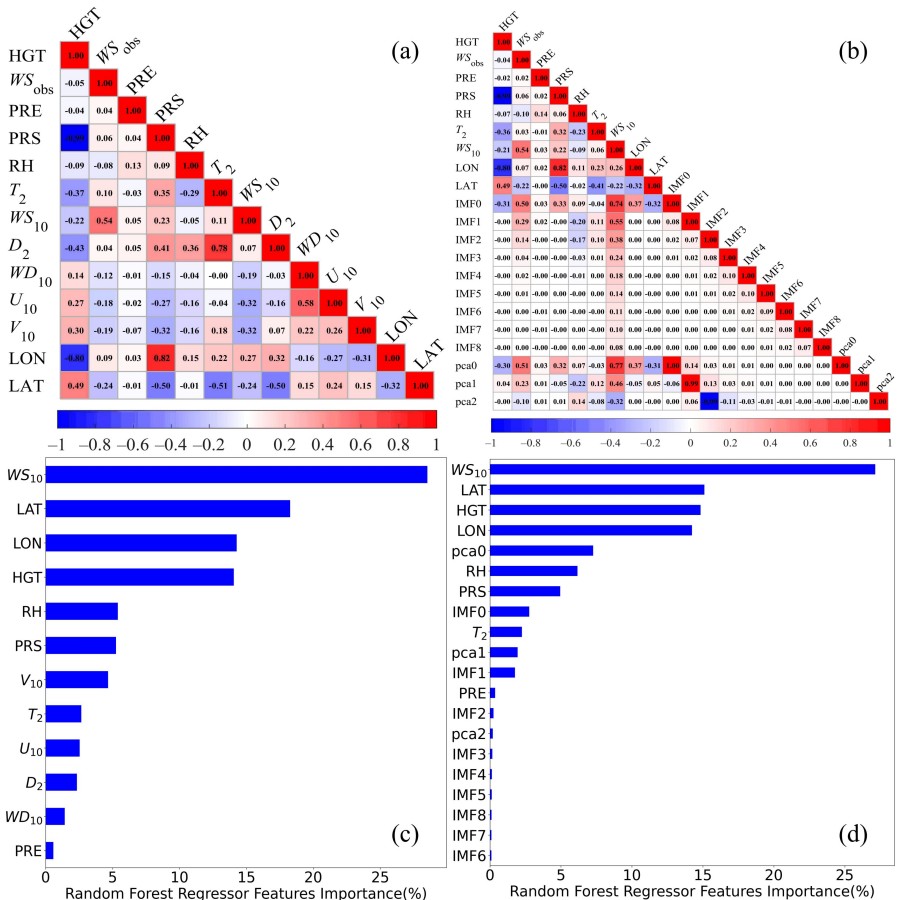

**Figure 6: Schematic diagram of correlation and feature importance for two sets of experiments. (a) and (c)**
**represent experiment 1, and (b) and (d) represent experiment 2.**



### 3.2 Experiment 2 evaluation


Experiment 2 builds upon Experiment 1, concentrating on the predicted 10-meter wind speed by
the WRF model. We use the VMD algorithm to decompose the predicted wind speed into 9
components, and use the PCA algorithm to extract the main 3 principal components. In the RF feature
importance analysis (Fig.6b, d), it is evident the VMD algorithm can decompose IMF0 and IMF1, with
contributions surpassing those of 2-meter temperature and precipitation, respectively. The importance
of the pca0 component, after PCA principal component extraction, reaches up to 8%. What is
particularly interesting is that in the correlation analysis, the correlation values between the IMF0 and
pca0 components and the actual wind speed are 0.50 and 0.51, which are second only to the forecasted
wind speed.
The RMSE between the predicted value and the observed value of the training set (validation set)
in the VMD-PCA-lightGBM, VMD-PCA-XGBoost, VMD-PCA-RF, VMD-PCA-DBN, and
VMD-PCA-MLP models are 0.33 m/s (0.53 m/s), 0.31 m/s (0.54 m/s), 0.52 m/s (0.57 m/s), 0.75 m/s
(0.75 m/s) and 0.60 m/s (0.66 m/s). The FA are 0.99 (0.94), 1.00 (0.94), 0.94 (0.93), 0.87 (0.87), and
0.91 (0.90). The R squared are 0.91 (0.77), 0.93 (0.77), 0.79 (0.73), 0.55 (0.55), and 0.71 (0.65). These
are shown in supplementary materials Figures S6-8. In comparison to the above five artificial
intelligence methods, training results of VMD-PCA-DBN are relatively inferior. VMD-PCA-lightGBM
and VMD-PCA-XGBoost models still train the processed data effectively. According to the scatter
density figure (Fig.7a, Fig.8a), the scatter points are relatively concentrated on the 1:1 line. The RMSE
of VMD-PCA-lightGBM, VMD-PCA-XGBoost, VMD-PCA-RF, VMD-PCA-DBN, and
VMD-PCA-MLP models on the test set in December 2021 (January 2022) are 0.63 m/s (0.63 m/s),
0.68 m/s (0.66 m/s), 0.62 m/s (0.64 m/s), 0.77 m/s (0.76 m/s), and 0.71 m/s (0.69 m/s) respectively.
The FA of the five models on the test set in December 2021 (January 2022) are 91.13 % (91.49 %),
89.22 % (90.23 %), 91.79 % (91.57 %), 87.93 % (87.61 %), and 87.20 % (88.94 %). The R are 0.81
(0.78), 0.78 (0.76), 0.82 (0.78), 0.71 (0.67), and 0.75 (0.73). The test results of the five models in
Experiment 2 in December 2021 and January 2022 show that the error indexes of RMSE and FA of
each model exhibit minimal difference in two months. Nonetheless, disregarding the correlation
coefficient (R) results, the performance of the five models in December 2021 is inferior to that in
January 2022. The diurnal variation scatter plot of two months is tested. The red scatter represents the



nighttime wind speed, which is more concentrated on the 1:1 line. In contrast, the blue scatter
represents the afternoon wind speed, which is slightly away from the 1:1 line. This suggests that the
correction effect of the five models exhibits a noticeable diurnal variation.

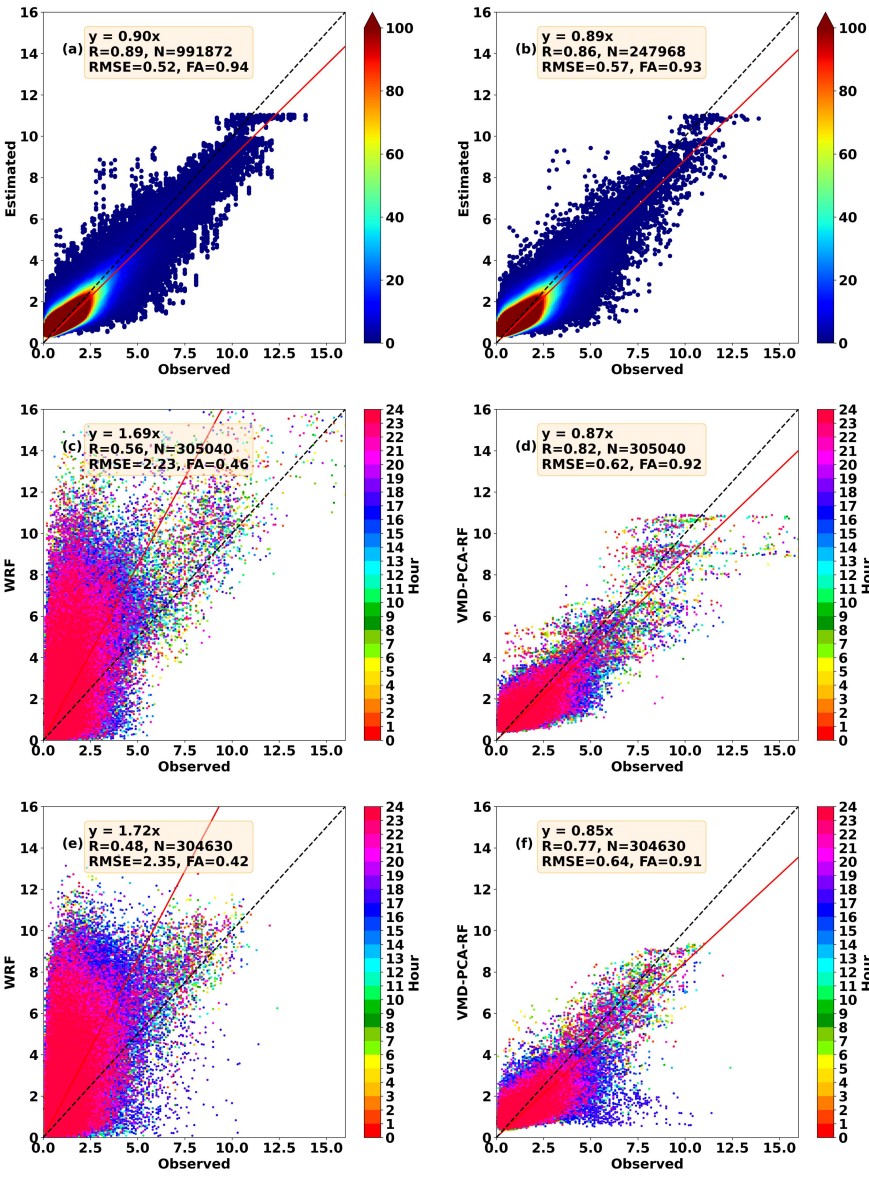


**Figure 7: The 24-hour scatter density map compared with the actual 10-meter wind speed. (a) 10-fold**
**cross-validation training set of VMD-PCA-RF model in February 2022, (b) 10-fold cross-validation**
**validation set of VMD-PCA-RF model in February 2022, (c) WRF forecasts in December 2021, (d)**



**VMD-PCA-RF model forecasts in December 2021, (e) WRF forecasts in January 2022, and (f)**
**VMD-PCA-RF model forecasts in January 2022.**



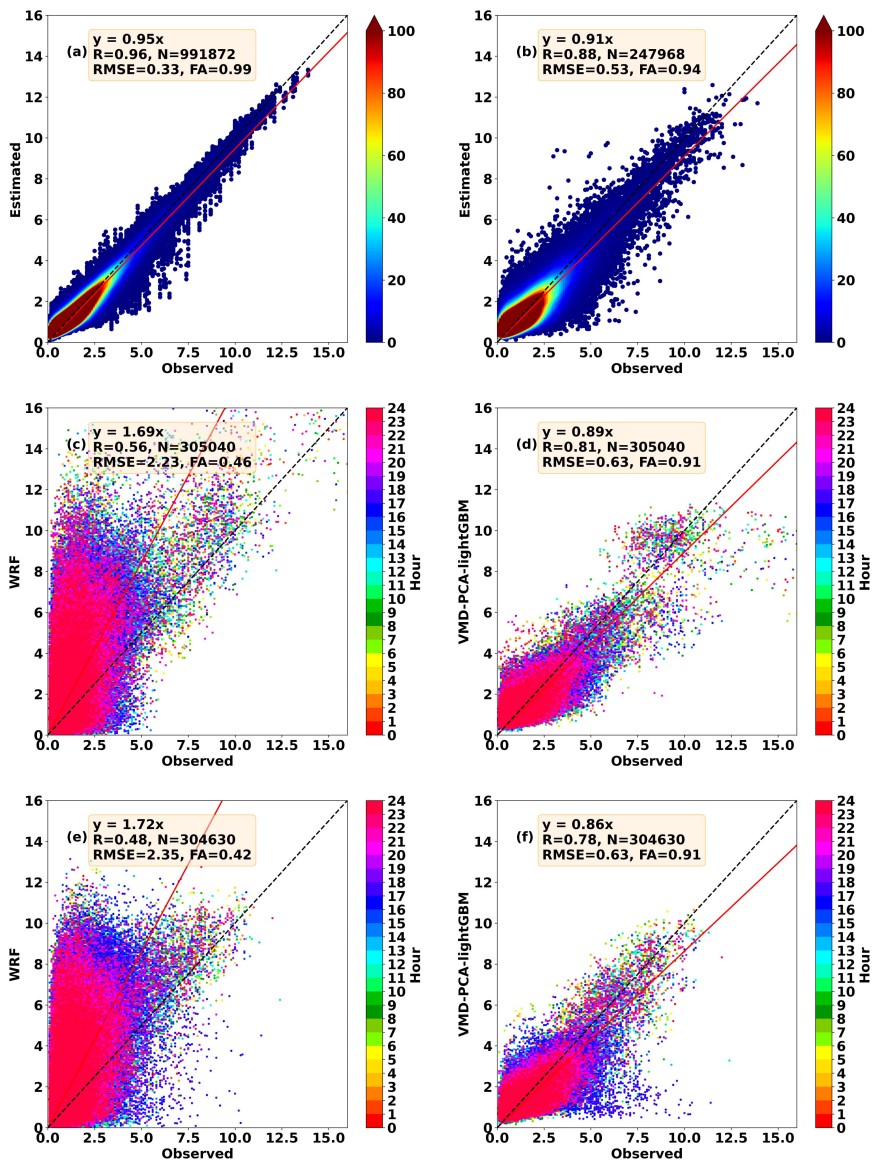

**Figure 8: The 24-hour scatter density map compared with the actual 10-meter wind speed. (a) 10-fold cross-validation training set of VMD-PCA-lightGBM model in February 2022, (b) 10-fold cross-validation validation set of VMD-PCA- lightGBM model in February 2022, (c) WRF forecasts in December 2021, (d) VMD-PCA- lightGBM model forecasts in December 2021, (e) WRF forecasts in January 2022, (f) VMD-PCA- lightGBM model forecasts in January 2022.**



**3.3 Comparison of the two experiments**


Firstly, all 10 models effectively corrected the 10-meter wind speed forecasted by WRF. Table 3
and Table 4 represent the evaluation indexes of wind speed errors predicted by 10 models in December
2021 and January 2022. From the two tables, it is evident that the VMD-PCA-RF and
VMD-PCA-lightGBM models have the best performance in December 2021 and January 2022,
respectively, with the most comprehensive performance of the forecast indicators. The MAE, RMSE,
rMAE, rMAE, and FA for the two models VMD-PCA-RF (VMD-PCA-lightGBM) were 0.46 m/s (0.45
m/s), 0.62 m/s (0.63 m/s), 37.36 % (34.75 %), 50.39 % (48.65 %), and 91.79 % (91.49 %) in December
2021 (January 2022). Additionally, based on the analysis of the Taylor chart (Fig.9e, f) of 10 models in
Fig.9, it can also be seen that the scatter distance of VMD-PCA-RF and VMD-PCA-lightGBM models
is closest to the observed black dotted line and the black triangle position. The two models show that
the standard deviation is close to the observed wind speed, with the lowest RMSE and the highest R.
Secondly, in the comparison of cumulative probability distributions, all models passed Kolmogorov's
5 % confidence interval test when the interval of wind speed is 0.5 m/s (Fig.9a, d). However, when the
interval of wind speed is 0.2 m/s (Fig.9b, e), VMD-PCA-lightGBM model deviated from
Kolmogorov's 5 % confidence interval detection in December 2021. This indicates that the
VMD-PCA-RF model has a better predictive effect than VMD-PCA-lightGBM model in December
2021 when the actual wind speed is within the range of 0.4 m/s-0.8 m/s.
**Table 3. Table of evaluation indexes of wind speed error predicted by 10 models in December 2021**

| Model | MAE(m/s) | RMSE(m/s) | rMAE(%) | rRMSE(%) | FA(%) | R |
|---|---|---|---|---|---|---|
| VMD-PCA-lightGBM | 0.47 | 0.63 | 37.67 | 51.25 | 91.13 | 0.81 |
| VMD-PCA-XGBoost | 0.49 | 0.68 | 39.84 | 54.82 | 89.22 | 0.78 |
| VMD-PCA-RF | 0.46 | 0.62 | 37.36 | 50.39 | 91.79 | 0.82 |
| VMD-PCA-DBN | 0.53 | 0.75 | 43.32 | 61.13 | 87.93 | 0.71 |
| VMD-PCA-MLP | 0.53 | 0.72 | 43.04 | 58.47 | 87.2 | 0.75 |
| lightGBM | 0.49 | 0.67 | 39.59 | 54.16 | 89.68 | 0.79 |
| XGBoost | 0.51 | 0.70 | 41.51 | 56.64 | 87.9 | 0.77 |
| RF | 0.48 | 0.65 | 38.80 | 52.32 | 90.64 | 0.81 |
| DBN | 0.56 | 0.77 | 45.25 | 62.46 | 86.74 | 0.71 |



| Model | MAE(m/s) | RMSE(m/s) | rMAE(%) | rRMSE(%) | FA(%) | R |
|---|---|---|---|---|---|---|
| MLP | 0.55 | 0.74 | 44.65 | 60.1 | 86.08 | 0.75 |


**Table 4. Table of evaluation indexes of wind speed error predicted by 10 models in January 2022**

| Model | MAE(m/s) | RMSE(m/s) | rMAE(%) | rRMSE(%) | FA(%) | R |
|---|---|---|---|---|---|---|
| VMD-PCA-lightGBM | 0.45 | 0.63 | 34.75 | 48.65 | 91.49 | 0.78 |
| VMD-PCA-XGBoost | 0.47 | 0.66 | 36.31 | 51.01 | 90.23 | 0.76 |
| VMD-PCA-RF | 0.46 | 0.64 | 35.06 | 49.00 | 91.57 | 0.78 |
| VMD-PCA-DBN | 0.53 | 0.75 | 40.96 | 57.49 | 87.61 | 0.67 |
| VMD-PCA-MLP | 0.50 | 0.69 | 38.46 | 53.16 | 88.94 | 0.73 |
| lightGBM | 0.46 | 0.64 | 35.24 | 49.34 | 91.11 | 0.77 |
| XGBoost | 0.48 | 0.67 | 36.68 | 51.38 | 89.88 | 0.75 |
| RF | 0.46 | 0.64 | 35.18 | 49.13 | 91.36 | 0.78 |
| DBN | 0.53 | 0.74 | 40.97 | 56.86 | 87.71 | 0.68 |
| MLP | 0.49 | 0.68 | 37.83 | 52.26 | 89.57 | 0.74 |


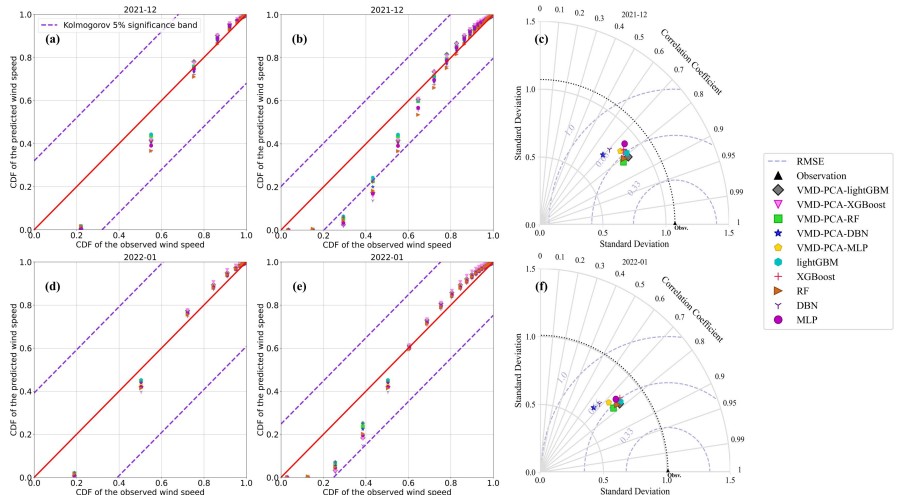


**Figure 9: The cumulative distribution probability scatter plots of the actual wind speed and the predicted**
**wind speed of 10 models in wind speed intervals of 0.5 m/s ((a) represents December 2021, (d) represents**
**January 2022) and 0.2 m/s ((b) represents December 2021, (e) represents January 2022) respectively; Taylor**
**distribution map ((c) represents December 2021, (f) represents January 2022).**





### 3.4 Spatial–temporal variations in the best models


Based on our comparative analysis results, we conclude that the best performing combination
models in December 2021 and January 2022 are VMD-PCA-RF and VMD-PCA-lightGBM
respectively. Figure 10 shows the diurnal variation corrections of the two best models for a given
month, as well as the diurnal variation of wind speed in the original WRF forecast. The wind speed of
the original WRF numerical weather forecast shows noticeable overestimation, which is confirmed in
Fig.8c and 8e. The scatter points of WRF forecast predominantly deviate towards the upper left corner,
with relatively low correlation coefficients, 0.56 and 0.23, respectively. Furthermore, the wind speed
forecast by WRF displays obvious diurnal variation traits, characterized by large errors between
afternoon and evening, specifically between 11:00 and 20:00 (Fig.10a, b). Moreover, the actual average
wind speed in January 2022 deviates from the range of one standard deviation of the WRF forecast
wind speed at 17:00 and 18:00. This demonstrates that the wind speed forecast by WRF is inaccurate
and exhibits substantial diurnal variation errors.
After the best model was corrected, the error of diurnal variation is significantly reduced (Fig.10c,
d). First, the average wind speed corrected by the best model is essentially consistent with the actual
average wind speed curve, with minimal error and no diurnal variation. Second, the one standard
deviation range of the corrected and actual wind speeds is also well-matched, indicating that the
corrected and actual wind speed distributions are consistent. The correction effect at 16:00 and 17:00
on January 2022 is suboptimal, which may be due to the insufficient generalization of the training
model and the excessive fluctuation of the actual wind speed at these two time points.
The FA (Fig.11a, b) and RMSE (Fig.11e, f) distribution of WRF forecast 10-meter wind speed at
410 stations in the five southern provinces shows that the 10-meter wind speed prediction effect of the
WRF model in Yunnan is superior to that in the other four provinces. In the Yunnan area, the FA of
most WRF forecast station 10-meter wind speeds exceeds 40 %, and RMSE value is mostly below 2.4
m/s. Conversely, in other regions, such as Guangxi, Guangdong and Hainan, the terrain is relatively flat.
The FA of the 10-meter wind speed forecast by WRF is as low as 30 % at some stations, and the
RMSE reaches up to 5.4 m/s. However, after the VMD-PCA-RF and VMD-PCA-lightGBM models are
corrected, the FA of most stations in the five southern provinces is as high as 90 %, and the RMSE is as
low as 0.6 m/s. Moreover, in Guangxi, Guangdong, and Hainan, where the WRF forecast effect is





subpar, the accuracy of the corrected 10-meter wind speed by VMD-PCA-RF (VMD-PCA-lightGBM)
is significantly improved.

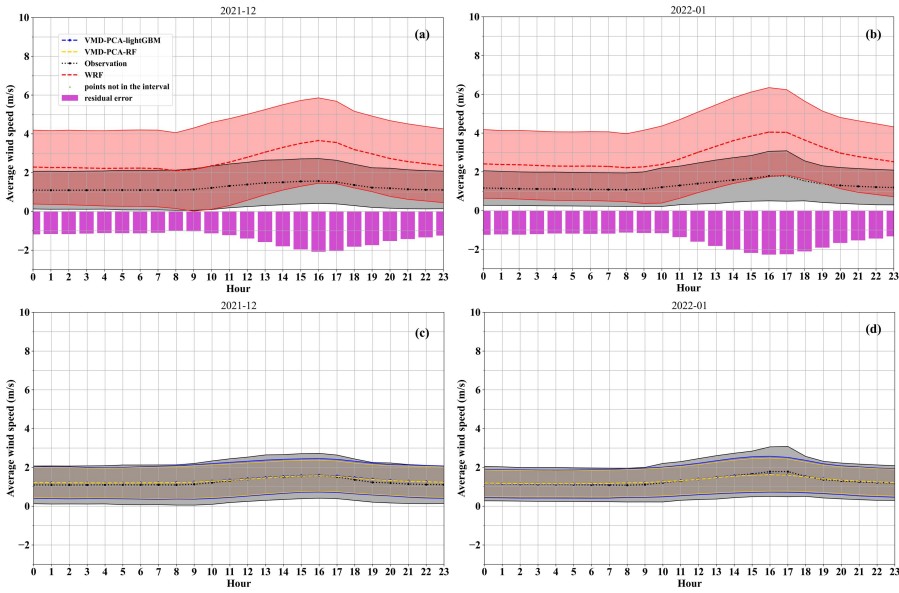

**Figure 10: VMD-PCA-lightGBM,VMD-PCA-RF and WRF daily variation of predicted and actual wind**
**speeds in December 2021 and January 2022.**

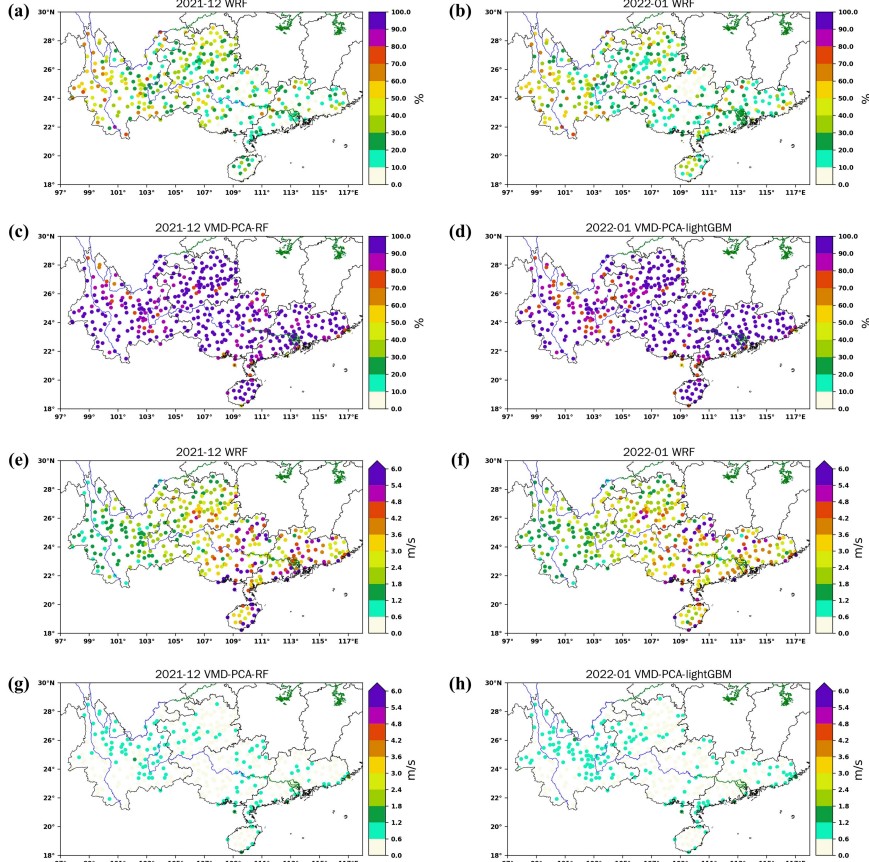

**Figure 11: FA and RMSE distribution maps of VMD-PCA-RF, VMD-PCA-lightGBM and WRF models on 410 sites in five southern provinces ((a), (c), (e), and (g) represent December 2021; (b), (d), (f), and (h) represent January 2022).**

**4. Discussion**

**4.1 The effects of BOA-VMD-PCA**

It is shown in Table 2 that the hyper-parameters of the 10 models in the two experiments are different. Since the DBN model is not added to the scikit-learn Python learning package, it is challenging to call the BOA algorithm for tuning parameters. Apart from the DBN model, all the other models are optimized using the BOA algorithm. From the various evaluation indicators in Table 3 and Table 4, the DBN model, which does not use the BOA algorithm to adjust the model parameters to





obtain an optimal parameter configuration, yields the worst prediction results in December 2021 and
January 2022. Moreover, studies (Xiong et al., 2022) also have shown that BOA can further improve
the model's prediction accuracy by configuring optimal hyper-parameters. The hyper-parameters such
as the number of neurons and learning rate in the hidden layer, significantly impact the model's
performance. When the same model is applied to different data sets of two experiments, the BOA
adaptively obtains the optimal combination of hyper-parameters, overcoming the limitations of manual
parameter adjustment (Guo et al., 2021). This suggests that the selection of model hyper-parameters
introduces considerable uncertainty into our prediction results. Therefore, the choice of optimization
model parameters represents one source of uncertainty in the correction results, which entails the
complexity of parameter selection. However, a more advanced parameter tuning method, such as the
BOA tuning algorithm, is essential.
The VMD is used to obtain unknown but meaningful features hidden in the 10-meter wind speed
sequences predicted using WRF models (Li et al., 2022). In addition, the PCA can extract important
components of anemometer subsequences. When the stationary subsequence serves as an input to the
error correction model, it contains more valuable information than the previous non-stationary wind
speed sequences (Xu et al., 2021).
The complexity of the input factors in this study is one of the sources of uncertainty in the process
of correcting WRF prediction results. The input factors of the two experiments are not identical. In the
second set of experiments, the input of meteorological factors is reduced based on the first set of
experiments, while component information of the 10-meter wind speed predicted by WRF is increased.
Multiple wind speed components processed by VMD-PCA and noise reduction are introduced. Among
them, the importance of pca0 and IMF0 introduced is approximately 5 %. In the 10-month test sets, the
correction accuracy of experiment 2 is no less than the results of experiment 1 (Fig.14, Fig.S9, 10),
indicating that the 10-meter wind speed components introduced by the VMD-PCA contribute
positively to the correction results.

**4.2 RF feature importance**

In order to further understand the feature importance ranking of the RF models, we divided the
model prediction results and actual wind speeds of the 410 stations into 20 equal parts according to



height (Fig.12). First of all, the actual wind speed in December 2021 and January 2022 varies with the
height of the station, showing that the lower the height of the station, the more significant the change of
wind speed. This relationship is associated with the wind speed profile of the atmosphere, where wind
speed increases as height decreases. Secondly, the wind speed during the day is generally greater than
the wind speed at night, which is related to the turbulent motion of the atmosphere during the day.
Solar radiation causes the atmosphere to mix, resulting in convective movement. The 10-meter wind
speed at night is affected by the cooling radiation of the surface, and the atmosphere is relatively stable.
The 10-meter wind speed predicted by WRF has the highest feature importance in the correction
process of the RF models. Input factors with distinct geographic information, such as latitude,
longitude, and height, rank highly in feature importance. Similarly, when Sun et al. 2019 used machine
learning to correct the 10-meter wind speed predicted by the numerical weather prediction model
ECMWF, the characteristic weight of the 10-meter wind speed predicted by the model is the highest,
followed by the sea-land factor. Also, as the 10-meter wind speed forecast by WRF increases, the
instability of the 10-meter wind speed corrected by the 10 machine learning models gradually increased,
and the correction accuracy gradually decreased (Fig.13). This partly explains the higher importance of
the 10-meter wind speed forecast by WRF.
With 1 km as the center, the measured 10-meter wind speed is more unstable in areas where the
station height increases or decreases. However, the 10-meter wind speed predicted by WRF being more
unstable with the station height decreases (Fig.12). The VMD-PCA-RF and VMD-PCA-lightGBM
models significantly reduce the instability of the 10-meter wind speed predicted by WRF. When the
height of the station increases or decreases at 1 km, the correction intensity tends to increase gradually.
This further explains the higher importance of the height factor in the RF model training.

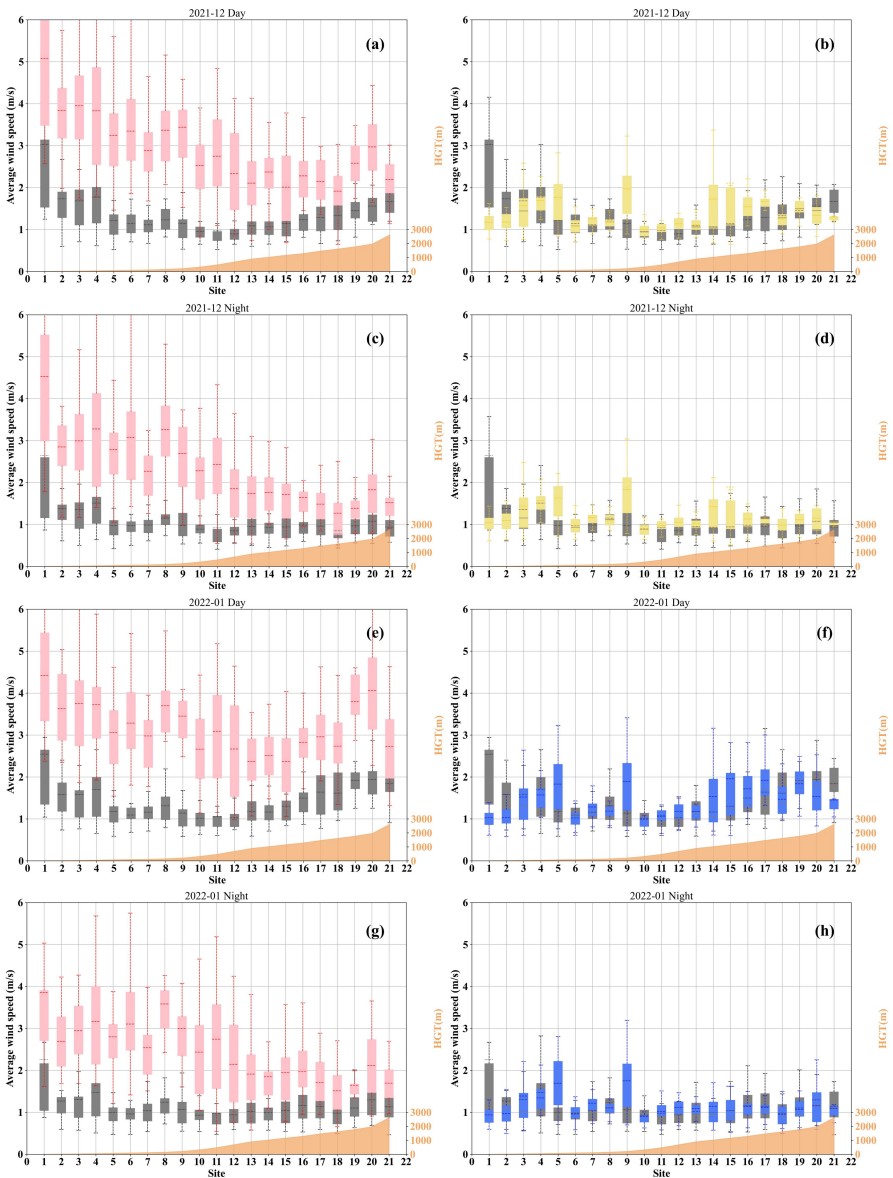

**Figure 12: The boxplots of the predicted wind speeds of the VMD-PCA-RF (yellow), VMD-PCA-lightGBM (blue), and WRF (pink) models at 20 stations at different height intervals, and the boxplots of the actual wind speeds (gray).**

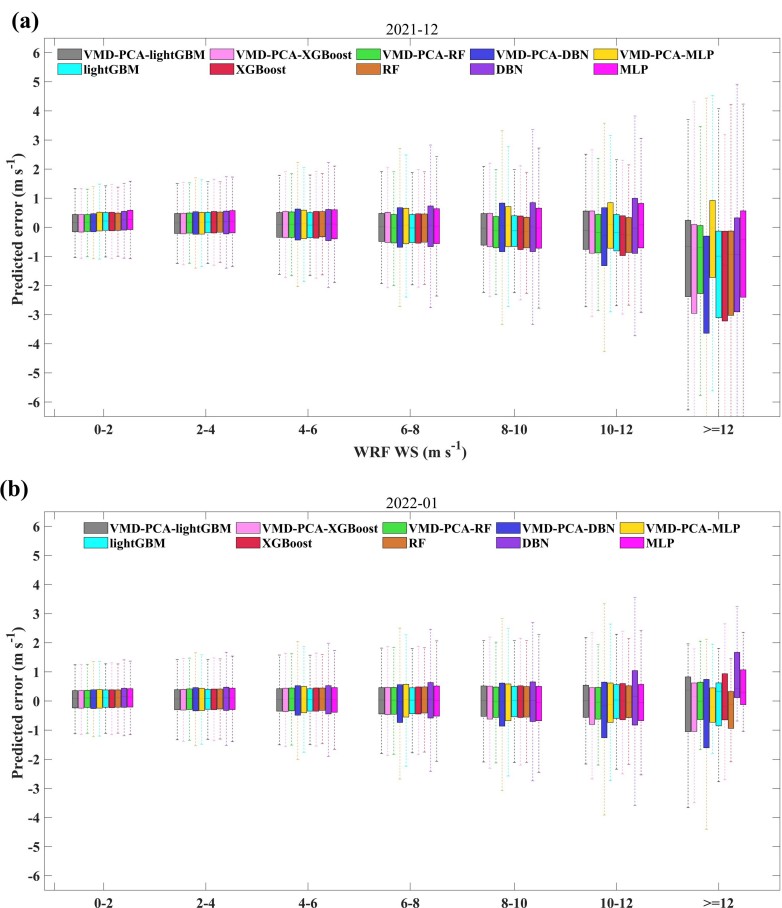

**Figure 13: The prediction error boxplots of 10 models in different WRF prediction intervals.**

### 4.3 Stability analysis of the proposed models

In order to identify the best model of the five southern provinces and assess the model's stability, we evaluated all 10 models over 10 different months. Fig.14 shows the evaluation histogram of the 10-meter wind speed predicted by the 10 models in Experiment 1 and Experiment 2, as well as the actual wind speed in various months. Meanwhile, Fig.S9 and S10 can more effectively illustrate the daily changes of the revised results of 10 models in 10 different months. As shown in the figure 14, the evaluation indexes of the model trained in Experiment 2, after VMD-PCA processing, outperform those of the model trained in Experiment 1. The RF model demonstrates exceptional robustness, while





the MLP model exhibits the poorest performance. VMD-PCA-RF evaluation indexes are relatively
stable across the 10 months, with a correlation coefficient R above 0.6, accuracy rate FA above 85 %,
MAE below 0.6 m/s, RMSE below 0.8 m/s, rMAE below 60 %, and rRMSE below 75 %. However, the
robustness of the VMD-PCA-lightGBM and VMD-PCA-XGBoost models is inferior to that of the
VMD-PCA-RF, with all six evaluation indexes performing worse than the VMD-PCA-RF as the
seasons and months change. In general, VMD-PCA-RF is the best wind speed correction model for
winter and even throughout the entire year in the five southern provinces.

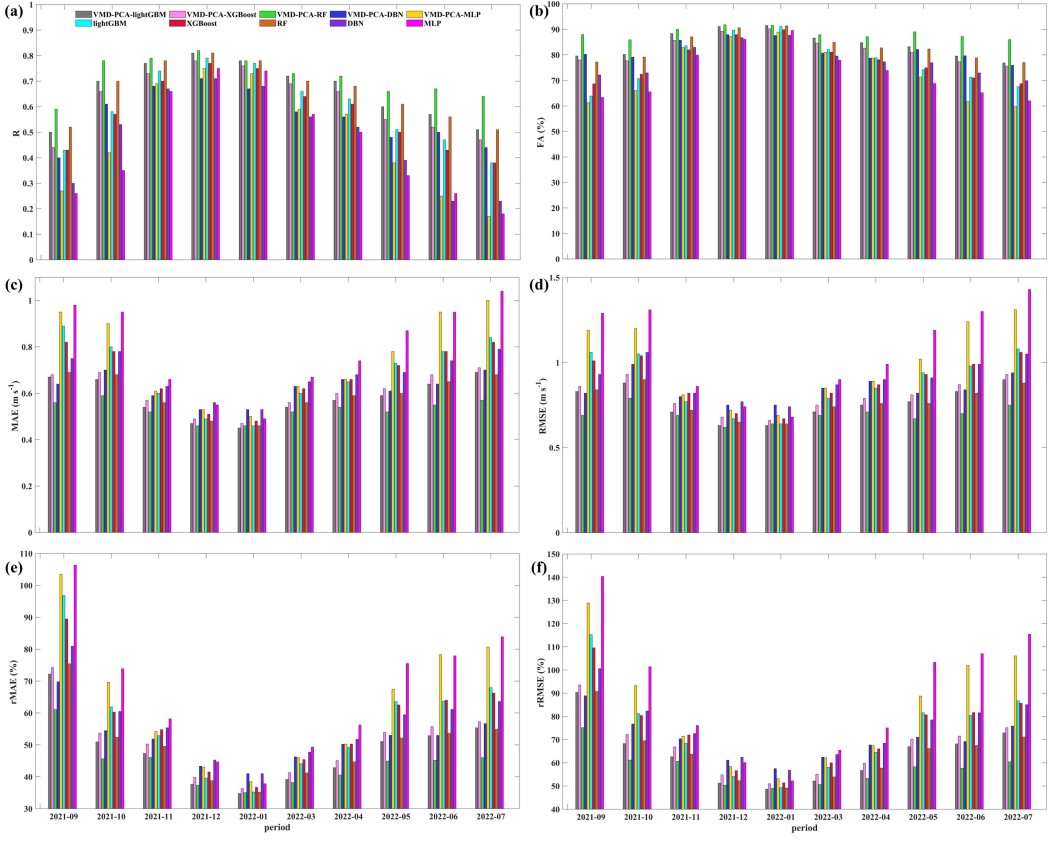


**Figure 14: Evaluation histograms of 10-meter wind speed predicted by 10 models and actual wind speed in**
**different months in Experiment 1 and Experiment 2 ((a), (b), (c), (d), (e), and (f) represent R, FA (%), MAE**
**(m/s), RMSE (m/s), rMAE (%), and rRMSE (%) respectively).**



**5. Conclusions**
In an effort to enhance the wind speed prediction performance for wind farms, this study
developed a WRF-based multi-step wind speed prediction model. A hybrid error correction strategy
combining BOA, VMD, PCA, and RF (LightGBM) is proposed to increase the accuracy of WRF
simulations. The first group of experiments used various meteorological elements as input factors in a
control experiment. In the second group of experiments, the wind speed sequence predicted by the
WRF model was decomposed into multiple IMFs using the VMD algorithm for feature extraction. A
principal component analysis method is used to extract meaningful principal components from these
subsequence IMFs to improve computational efficiency. In the error correction model, RF (lightGBM)
and other algorithms are used to train the relationship between different input factors and the actual
wind speed error, respectively.
Through a case analysis of 410 stations in five southern provinces in China, the following
conclusions can be drawn: (1) The machine learning models tuned by the BOA-VMD-PCA algorithm
exhibit a positive impact on wind speed error correction; (2) Feature importance analysis revealed that
the top eight contributing factors for correcting WRF forecasted wind speed include WRF forecast
10-meter wind speed (WS10), latitude, longitude, altitude, pca0, humidity, pressure, IMF0; (3)
VMD-PCA-RF and VMD-PCA-lightGBM are the most suitable wind speed correction algorithms for
December 2021 and January 2022, respectively. The MAE, RMSE, FA, rMAE, rRMSE, and R of the
corrected wind speed and the actual wind speed are 0.46 (0.45), 0.62 m/s (0.63 m/s), 37.36 %
(34.75 %), 50.39 % (48.65 %), 91.79 % (91.49 %), and 0.82 (0.78); and (4) The proposed wind speed
correction model (VMD-PCA-RF) demonstrates the highest prediction accuracy and stability in the
five southern provinces in nearly a year and at different heights. VMD-PCA-RF evaluation indexes for
10 months remain relatively stable: correlation coefficient R is above 0.6, accuracy rate FA is above
85 %, MAE is below 0.6 m/s, RMSE is below 0.8 m/s, rMAE is below 60 %, and rRMSE is below
75 %. In future research, the proposed VMD-PCA-RF algorithm can be extrapolated to the 3 km grid
points of the five southern provinces to generate a 3km grid-corrected wind speed product.




**Code availability**

The code and model are available as a free-access repository on Zenodo at
https://doi.org/10.5281/zenodo.7940686 (Zhou, 2023).

**Data Availability**

The data is available as a free-access repository on Zenodo at https://doi.org/10.5281/zenodo.7940686
(Zhou, 2023).

**Author contributions**

SZ developed the software, visualized the data, and prepared the original draft. SZ and YG developed
the methodology and carried out the formal analysis. XX and SZ validated data. SZ, YG, XX, ZD, and
YL reviewed and edited the text. All authors have read and agreed to the published version of the
paper.

**Competing interests**

The authors declare that they have no conflict of interest.

**Financial support**

This research has been supported by the second batch of service public bidding projects for EHV
transmission companies in 2022 (2022-FW-2-ZB) (grant no. CG0100022001526556).



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
