# Peer review of "A robust error correction method for numerical weather"

_EGUsphere, 2023_

## Author Comment (AC1)

**Response to reviewer comments**

Dear Editor and Reviewer,

We are very grateful for your time and valuable comments, which we found very helpful. We have addressed questions and comments raised by the reviewer in the revised manuscript with tracked changes. Please find our point-by-point response (in blue font) to the comments below. We hope our revisions have properly addressed your concerns.

Thanks again for your time.

Sincerely,
The authors

**Reviewer 1**

In this paper, the authors present interesting methods for wind speed corrections from the NWP model with multi-step methods. Below are a few minor suggestions for revision:

1. The main issue that I see in this paper is the short period for training and testing of the model, and the authors claim from this that the model is robust. Similar studies for wind speed correction from NWP models usually use several years for training and at least one year for testing. As I understood, this paper is trained only on data from February 2022, and the main conclusions are based on testing in December 2021 and January 2022, with some additional verification of stability over 10 months.

**Response:** Many thanks for pointing this out. While it is true that similar past studies for wind speed correction from NWP models usually use several years for training and at least one year for testing and our periods are shorter, the size of our data set is sufficient, if not greater than others'. For example, Sun et al. (2019) used a data set that contained 1827 days, from January 2012 to December 2016, using 143 grid points with a resolution of 0.5°*0.5° predicted by ECMWF, followed by 24 features for each sample, with a training set size of 1827*143*24 for each prediction time. Meanwhile, the size of our training set mentioned in lines 238-242 is about 2160*410*12. Therefore, even though it only took us a month to train, we actually trained millions of data; Second, the training data we used was obtained through daily operational runs of numerical weather forecasting, so we would have to run it for several years to get an equal amount of training data. The data we tested were mainly used to analyze the spatiotemporal changes after the model revision in December 2021 and January 2022. All the indicators of the proposed model (VMD-PCA-RF) are relatively robust for the other eight months. We will continue to add new training datasets going forward, however, it will be a challenge to train data over several million levels.

2. order of figures in the text: Fig. 1, Fig. 2, Fig. 3, Fig. 6, Fig. 4, Fig. 5, ... Fig. 11,

Fig. 14, Fig. 12.

**Response:** Many thanks for your suggestion. We have adjusted the order of the figures.

3. Sometimes authors refer to figures in the text as "Fig. NN" in other cases as "Figure NN", and even once as "figure NN". According to Journal rules, I think it should always be "Fig. NN." Fig. 6 and 9 are unreadable.

**Response:** Many thanks for your suggestion. We have corrected them all to "Fig. NN".

4. On lines 56–57, the authors state that "Currently,..." and cite a publication from 1999, but there are more recent publications for the HIRLAM model or consortium.

**Response:** Many thanks for your suggestion. We have updated to a more recent reference.

5. The authors claim in line 520 that "In general, VMD-PCA-RF is the best wind speed correction model for winter and even throughout the entire year in the five southern provinces," while on Fig. 14 for 2022-01, VMD-PCA-lightGBM is better.

**Response:** Thank you very much for pointing this out. As seen in Table 4, although VMD-PCA-lightGBM model has the best indicators for January 2022, compared to VMD-PCA-RF, the errors of the two models in various indicators are very small, and the error of MAE and RMSE is only 0.01 m/s. However, in Fig.14, the VMD-PCA-lightGBM model performed worse than VMD-PCA-RF in all of the other 9 months except January 2022.

To clarify this, we have added the following in the text: "In general, VMD-PCA-lightGBM is the superior wind speed correction model for the winter, and VMD-PCA-RF performs the best throughout the entire year in the five southern provinces."

6. There should be more clarification about observational data. In line 132, the authors wrote "For the purposes of this paper, the 10-meter wind speed data is interpolated across 410 sites". Are those 410 sites the weather stations? Why did the authors use interpolation from this database instead of observations from stations?

**Response:** Thank you for drawing our attention to this. The observed data comes from the China Meteorological Administration land data assimilation system (CLDAS-V2.0) real-time product data set (https://data.cma.cn/data/cdcdetail/dataCode/NAFP_CLDAS2.0_RT.html).

After post-processing by the China Meteorological Public Service Center, the data's resolution is reduced to 3km by 3km, and it is interpolated into the meteorological station. The observed data source has been integrated with the observation data of weather stations for consistency.

To clarify this and add more context to the data description, we have added the following in the text: "The observed data comes from the China Meteorological Administration land data assimilation system (CLDAS-V2.0) real-time product data

set
(https://data.cma.cn/data/cdcdetail/dataCode/NAFP_CLDAS2.0_RT.html).
After post-processing by the China Meteorological Public Service Center, the data's resolution is reduced to 3km by 3km, and it is interpolated into the meteorological station. The observed data source has been integrated with the observation data of weather stations for consistency."

---

## Author Comment (AC2)

**Response to reviewer comments**

Dear Editor and Reviewer,

We are very grateful for your time and valuable comments, which we found very helpful. We have addressed questions and comments raised by the reviewer in the revised manuscript with tracked changes. Please find our point-by-point response (in blue font) to the comments below. We hope our revisions have properly addressed your concerns.

Thanks again for your time.

Sincerely, The authors

**Reviewer 2**

Zhou et al. present a series of machine learning models including VMD-PCA-RF, a combination of Variational Mode Decomposition, Principal Component Analysis, and Random Forest, for correction of errors in WRF-predicted wind speeds. The manuscript presents various machine learning algorithms and uses two sets of experiments with different approaches to arrive at the model with better predictive capabilities. Accurate prediction of wind speed is important for the wind energy market for effective harvesting of wind energy, and this manuscript has potential in improving predictive capabilities for such uses. As a modeling paper it is fit for the scope of GMD, but the manuscript as presented has major shortcomings primarily in its presentation that require major revisions before further considation.

Major comments:

- Many of the figures in the text are unclear both in presentation and in purpose. Generally, the use of figures to illustrate points and their order in the text should be deliberate and help the flow of the reader to understand the text.

For example, figure 1 shows the elevation map of five southern provinces in China where observational data is used. Figure 2 shows the WRF simulation domain which appears to be a direct figure output from the WRF Pre-Processor (WPS). What is the purpose of these figures? It could be merged into one figure where the observation sites and provinces are marked. The purpose of the elevation maps in the analysis only shows up very late in the text in Section 4.2 about the RF feature importance and is not immediately clear to the reader.

**Response:** Many thanks for your suggestion. Fig. 1 shows the observation data evenly distributed in the five southern provinces, with the purpose of introducing specific locations of the observation data. The purpose of Fig. 2 is to illustrate the scope of WRF nesting regions. We took your advice and have combined the two figure. The combined figure is shown in Fig. 1\* below and replaced in the manuscript.

---

## Author Comment (AC3)

<h1 style="text-align:center">Response to reviewer comments</h1>

Dear Editor and Reviewer,

We are very grateful for your time and valuable comments, which we found very helpful. We have addressed questions and comments raised by the reviewer in the revised manuscript with tracked changes. Please find our point-by-point response (in blue font) to the comments below. We hope our revisions have properly addressed your concerns.

Thanks again for your time.

Sincerely,
The authors

**Reviewer 3**

General comments:

This paper is a description of several candidate post-processing approaches for producing point wind speed forecasts from numerical weather prediction simulations for sites in southern China. While the overall methods appear reasonable, and the conclusions appear valid, the paper needs some work to clarify the approach in some regards.

**Response:** Thank you very much for your recognition and encouragement of our work, we will further modify it according to your comments.

Specific comments:

1. I think more detail is needed on the gridded meteorological dataset that you describe creating in section 2.1 "Data".  In particular, what is the source of the meteorological in situ observations?  Are these wind towers all at a consistent height? How do you combine the surface observations with satellite data?  Can you show some proof that your dataset "exhibits superior quality compared to other products", or at least provide some references that evaluate the dataset?

**Response:** Thank you very much for pointing this out. The observed data comes from the China Meteorological Administration land data assimilation system (CLDAS-V2.0) real-time product data set (https://data.cma.cn/data/cdcdetail/dataCode/NAFP_CLDAS2.0_RT.html).

These data are processed by the China Meteorological Public Service Center to equivalent latitude and longitude grid scale, covering a geographical range of 15-32.97°N and 94-120.97°E. The spatial resolution of the grid is 0.03° × 0.03° (3km by 3km) and the temporal resolution is 1 hour. China Meteorological Public Service Center applied the nearest neighbor interpolation for precipitation, and bilinear interpolation for the other four meteorological elements with downscaling from 3km to 410 sites.

Yes! These actual observed wind speed data are obtained from the meteorological

station location at a height of 10 m.

"Combining the surface observations with satellite data" and "exhibits superior quality compared to other products" are CLDAS-V2.0 official website documentation description results.

To clarify this and we have rewritten the Data part into the following text: "*The observed data comes from the China Meteorological Administration land data assimilation system (CLDAS-V2.0) real-time product data set (https://data.cma.cn/data/cdcdetail/dataCode/NAFP_CLDAS2.0_RT.html).*

*According to the description of the documents on the official website, the dataset is constructed through the integration of multiple sources, including ground and satellite data, and is refined using advanced techniques such as multi-grid variational assimilation, physical inversion, and terrain correction. This dataset exhibits superior quality in comparison to other products, offering higher spatial and temporal resolutions.*

*The target observation data includes 2-m air temperature, 2-m specific humidity, 10-meter wind speed, surface pressure, and precipitation. These data are processed by the China Meteorological Public Service Center to equivalent latitude and longitude grid scale, covering a geographical range of 15-32.97°N and 94-120.97°E. The spatial resolution of the grid is 0.03° × 0.03° (3km by 3km) and the temporal resolution is 1 hour.*

*China Meteorological Public Service Center applied the nearest neighbor interpolation for precipitation, and bilinear interpolation for the other four meteorological elements with downscaling from 3km to 410 sites. We selected the 10-meter wind speed data of 410 sites, as illustrated in Fig. 1.*"

2. In the step 1 description (first part of Fig. 3, and text description in lines 222-242), it seems like a lot is being changed between Exp. 1 and Exp. 2.   How can you control for this?   It makes it somewhat hard to interpret the results.

**Response:** Thank you very much for pointing this out. In terms of controlled experiments, the control of variables in our two experiments is not very strict. However, on the basis of experiment 1, experiment 2 eliminated four meteorological variables whose feature importance was less than 4 %, which were called $U_{10}$, $V_{10}$, $D_2$, and $WD_{10}$. Most importantly, when we introduced the VMD-PCA algorithm, the feature importance of pca0 and IMF0 both exceeded 5 %. In other words, we retained the most important meteorological variable for correcting forecast wind speed in experiment 1, and introduced pca0 and IMF0 wind speed sub-series processed by VMD-PCA algorithm.

3. I was confused about why no bias statistics were shown in the verification section. Showing only mean absolute error and root mean square error type verification is only part of the story;   can you say anything about the mean biases of the different approaches explored in this study?     I think that is an important part of the analysis

that is not shown yet.

**Response:** Thank you very much for your question. I have read some literature (Xiong et al., 2022; Zhang et al., 2019; Xu et al., 2021) about correcting wind speed forecasting, and I hardly saw the mean biases as a statistical indicator. According to our understanding, if at some time point the predicted wind speed is higher than the actual wind speed, then its bias is a positive value such as +x. If at other time point the predicted wind speed is lower than the actual wind speed, then its bias is negative such as -x. In this case, the calculated mean biases may be 0, which is not very suitable for evaluating the forecast of wind speed.

Xiong, X., Guo, X., Zeng, P., Zou, R., and Wang, X.: A Short-Term Wind Power Forecast Method via XGBoost Hyper-Parameters Optimization, Front. Energy Res., 10, 905155, https://doi.org/10.3389/fenrg.2022.905155, 2022.

Xu, W., Liu, P., Cheng, L., Zhou, Y., Xia, Q., Gong, Y., and Liu, Y.: Multi-step wind speed prediction by combining a WRF simulation and an error correction strategy, Renewable Energy, 163, 772–782, https://doi.org/10.1016/j.renene.2020.09.032, 2021.

Zhang, Y., Chen, B., Pan, G., and Zhao, Y.: A novel hybrid model based on VMD-WT and PCA-BP-RBF neural network for short-term wind speed forecasting, Energy Conversion and Management, 195, 180–197, https://doi.org/10.1016/j.enconman.2019.05.005, 2019.

4. I think somewhere (maybe in Fig. 1 and/or Fig. 2) you need to label the provinces, as readers from outside of China may not know which is which.

**Response:** Many thanks for your suggestion. We have labeled the provinces in Fig. 1. The revised figure is shown in Fig. 1 below.

[Figure]

**Figure 1: WRF model simulation area elevation diagram. (d02 represents the nested area of the second layer of the WRF model, and the black triangles represent the meteorological sites).**

5. Minor comments:

Page 1, line 24: GFS stands for Global Forecast System.

**Response:** Thank you very much for pointing this out. We have corrected line 24 to the following text: "*We first construct WRF-predicted wind speeds using the Global Forecast System (GFS) model output based on prediction results.*"

6. Lines 29: indexes > indices.

**Response:** Thank you very much for pointing this out. We have corrected line 29 to the following text: "*We find that the VMD-PCA-RF evaluation indices exhibit relative stability over nearly a year:*"

7. Page 3, line 83: training the > training on the

**Response:** Thank you very much for pointing this out. We have corrected line 83 to the following text: "*The error correction model improves the accuracy of the NWP model by training on the relationship between the NWP predictor variables and the observed correlation variables.*"

8. Page 5, line 118: Can you specify that these provinces are in China?

**Response:** Thank you very much for pointing this out. We have corrected line 118 to the following text: "*We analyze six distinct wind speed error indicators to compare and identify the most suitable wind speed error correction schemes for five southern provinces (Yunnan, Guizhou, Guangxi, Guangdong, Hainan) in winter and throughout most of the year.*"

8. Page 6, line 141: NCEP does not develop WRF, but rather NCAR (National Center for Atmospheric Research).    While there are contributors to WRF from NCEP, there are also contributors from universities and many other organisations.

**Response:** Thank you very much for pointing this out. We've corrected line 140-141 to the following text: "*The WRF 4.2 model, developed by the National Center for Atmospheric Research (NCAR), ...*"

9. Page 6, lines 144-147: This section about GFS is confusing. Are you saying WRF uses GFS initial and lateral boundary conditions? It has the capability, but is not required to use GFS data. Also, NCAR did not have a role in developing GFS to my knowledge.

**Response:** Thank you very much for pointing this out. We've updated section 2.2.1 to the following text:

"*The WRF 4.2 model, developed by the National Center for Atmospheric Research (NCAR), represents a new generation of mesoscale numerical models with numerous applications in research forecasting. When forecasting meteorological elements, the WRF model normally uses the GFS data developed by the National Center for Environmental Forecasting (NCEP). Using the WRF model in combination with daily GFS data resolution of 0.25° × 0.25°, the GFS data updates at 06:00 UTC and generates forecasting every 3 hours for a total duration of 90 hours. We selected the 24-h forecasting data from the WRF-resulted file after spin-up time of 18 hours. The GFS data as the initial field and lateral boundary conditions for the WRF model. Surface static data, such as terrain, soil data, and vegetation coverage, are derived from the Moderate Resolution Imaging Spectroradiometer (MODIS) satellite with a resolution of 15 seconds (approximately 500 meters). Incorporating a two-layer grid nesting configuration, the forecast area is illustrated in Fig. 1. The WRF configuration process is detailed in Table 1. Given that the time scale of the meteorological station data in the study area is 1 hour, the forecast data time interval of the WRF model is also set to 1 hour. As a widely used numerical weather forecast model, the WRF model is suitable for weather studies from a few meters to several thousand kilometers. Therefore, this paper uses the WRF model to predict 10-meter wind speed as the input factor for the error correction model (Xu et al., 2021).*"

10. Page 7, line 166: surface process plan > land surface model.

**Response:** Thank you very much for pointing this out. We have deleted this content according to Reviewer 2's opinion.

11. Page 9, line 206: Can you define and capitalize your acronym "pcs".

**Response:** Thank you very much for pointing this out. It means principal components (Pcs). We've corrected line 206 to the following text: "*When principal components (Pcs) are used as the input of the error prediction algorithm, the Pcs fully reflect the characteristics of the subsequence and reduce the model complexity.*"

12. Fig. 3: validing > validating.

**Response:** Thank you very much for pointing this out. The revised figure is shown in Fig. 2 below.

[Figure]

**Figure 2: Flowchart of the AI model used to correct WRF-predicted wind speeds in the two main experimental pathways.**

13. Page 10, line 228: selected WRF field forecast data, including > selected WRF field forecast data to include only…

**Response:** Thank you very much for pointing this out. We've corrected line 228 to the following text: "*Experiment 2, as illustrated in Fig. 6(d), derives 8 sets of data by reducing the selected WRF field forecast data, including only altitude, 10-meter wind speed, latitude, longitude, surface pressure, relative humidity, 2-meter temperature, and hourly precipitation.*"

14. Page 10, line 235: 8+9+3 does not equal 12. Are you counting the 9 IMF components as one set of meteorological elements? Please clarify your wording here.

**Response:** Thank you very much for pointing this out. we have corrected line 235 to the following text: "*Experiment 1 (Experiment 2) standardize 12 sets of meteorological elements (8 sets of meteorological elements in Fig. 2, 9 IMF components, and three PCA vectors in Fig. 4) and wind speed observation data, respectively.*"

[Figure]

**Figure 4: Three-dimensional view of 12 wind speed components after VMD and PCA processing of the 10-meter forecast wind speed at Lechang Station in Guangdong from December 1, 2021, to February 28, 2022.**

15. Page 13, line 276: Where does the "FA" acronym come from? I normally interpret that as false alarm, but it seems you have a different definition.

**Response:** Thank you very much for pointing this out. It means Forecasting Accuracy

(FA), which has been used in past literature (Sun et al., 2019).

Sun, Q., Jiao, R., Xia, J., Yan, Z., Li, H., Sun, J., Wang, L., and Liang, Z.: Adjusting Wind Speed Prediction of Numerical Weather Forecast Model Based on Machine Learning Methods. Meteorological Monthly, 45(3): 426-436. https://doi.org/10.7519/j.issn.1000-0526.2019.03.012, 2019.

16. Line 278: index > indices.
**Response:** Thank you very much for pointing this out. We have corrected line 278 to the following text: "*The formula for calculating the error indices is as follows:*"

17. Lines 303-308: Can you put these verification results in a Table? That would make it much easier to read, and to compare the different approaches. The same goes for further lists of results in other sections.
**Response:** Thank you very much for pointing this out.
Sure! The testing set results in lines 303-308 are shown in Tables 3 and 4 of the original paper. As for the results of training and verification, as mentioned by Reviewer 2, we put the results in Table 3*.
Of course, for clarity, we have added Table 3* to show the error indices of the training set and validation set of the 10 AI models in two sets of experiments in February 2022.

**Table 3*. Table of evaluation indices of wind speed error trained and verified by 10 models in February 2022**

| Model | training set | | | validation set | | |
|---|---|---|---|---|---|---|
| | R | RMSE(m/s) | FA | R | RMSE(m/s) | FA |
| VMD-PCA-lightGBM | 0.96 | 0.33 | 0.99 | 0.88 | 0.53 | 0.94 |
| VMD-PCA-XGBoost | 0.96 | 0.31 | 1.00 | 0.87 | 0.54 | 0.94 |
| VMD-PCA-RF | 0.89 | 0.52 | 0.94 | 0.86 | 0.57 | 0.93 |
| VMD-PCA-DBN | 0.74 | 0.75 | 0.87 | 0.74 | 0.75 | 0.87 |
| VMD-PCA-MLP | 0.84 | 0.60 | 0.91 | 0.81 | 0.66 | 0.90 |
| lightGBM | 0.93 | 0.41 | 0.98 | 0.88 | 0.54 | 0.94 |
| XGBoost | 0.96 | 0.31 | 0.99 | 0.87 | 0.56 | 0.93 |
| RF | 0.89 | 0.52 | 0.94 | 0.86 | 0.57 | 0.93 |
| DBN | 0.76 | 0.73 | 0.88 | 0.76 | 0.73 | 0.88 |
| MLP | 0.85 | 0.59 | 0.92 | 0.83 | 0.62 | 0.91 |

18. Line 309: Indexes > indices.

**Response:** Many thanks for your suggestion. We've corrected Line 309 to the following text: "*Considering different evaluation indices,*"

19. Lines 309-311: These sentences don't make much sense. It would be better to say "in" instead of "is that". For example FA in January 2022 is generally higher than in December 2021.

**Response:** Thank you very much for pointing this out. We've corrected Line 309-311 to the following text: "*Considering different evaluation indices, the revision effects of the five models in two months demonstrate that RMSE in January 2022 is generally lower than December 2021; FA in January 2022 is generally higher than December 2021; R in January 2022 is generally lower than December 2021.*"

20. Lines 356-359: Can you point out which figure this text refers to? I see some panels in Figs. 7 and 8 have blue and red scatter plots. Which model(s) are you specifically referring to about the day vs. night issues?

**Response:** Thank you very much for your question. It refers to Fig. S6d, f, Fig. S7d, f, Fig. S8d, f, Fig. 7d, f and Fig. 8d, f. We've corrected Line 309 to the following text: "*As is shown in Fig. S6d, f, Fig. S7d, f, Fig. S8d, f, Fig. 7d, f and Fig. 8d, f, the red scatter represents the nighttime wind speed, which is more concentrated on the 1:1 line. In contrast, the blue scatter represents the afternoon wind speed, which is slightly away from the 1:1 line. This suggests that the correction effect of the five models (VMD-PCA-lightGBM, VMD-PCA-XGBoost, VMD-PCA-RF, VMD-PCA-DBN, and VMD-PCA-MLP) exhibits a noticeable diurnal variation.*"

21. Figs. 7 and 8: Please clarify in the caption that the scatter plots are by hour. Is there some pattern to the models and months that are being shown in each panel? If so, it is above my head. Also, what is the difference between Fig. 7 and Fig. 8?

**Response:** Thank you very much for pointing this out.

For clarity, we have refined the headings of Fig. 7 and Fig. 8. The difference between Fig. 7 and Fig. 8 is the result of two different models, VMD-PCA-RF, VMD-PCA-lightGBM, respectively.

For example, we have corrected the title of Fig.7 to the following text: "*Figure 7: The scatter density map compared with the actual 10-meter wind speed: (a) 10-fold cross-validation training set of VMD-PCA-RF model in February 2022, (b) 10-fold cross-validation validation set of VMD-PCA-RF model in February 2022.*

*The 24-hour scatter map compared with the actual 10-meter wind speed: (c) WRF forecasts in December 2021, (d) VMD-PCA-RF model forecasts in December 2021, (e) WRF forecasts in January 2022, and (f) VMD-PCA-RF model forecasts in January 2022.*"

22. Fig. 11: Can you clarify in the caption which panels show FA and which show RMSE? It is not clear.

**Response:** Thank you very much for pointing this out.

For clarity, we have corrected the title of Fig. 11 to the following text: "*Figure 11: FA ((a), (b), (c), and (d)) and RMSE ((e), (f), (g), and (h)) distribution maps of VMD-PCA-RF, VMD-PCA-lightGBM and WRF models on 410 sites in five southern provinces ((a), (c), (e), and (g) represent December 2021; (b), (d), (f), and (h) represent January 2022)*."

23. Line 477: I think it would be clearer to say "elevation above sea level" rather than "height". When I read "height" in this sort of study, it makes me think of anemometer height above ground level.
**Response:** Thank you very much for pointing this out.
For clarity, we have corrected Line 477 to the following text: "*In order to further understand the feature importance ranking of the RF models, we divided the model prediction results and actual wind speeds of the 410 stations into 20 equal parts according to terrain height above sea level (Fig.12).*"

23. Line 494-495: This sentence is poorly worded and doesn't make sense.
**Response:** Thank you very much for pointing this out.
For clarity, we have corrected Line 494-495 to the following text: "*With 1 km as the center, the measured 10-meter wind speed is more unstable in areas where the station terrain height increases or decreases. However, the pink box of the 10-meter wind speed predicted by WRF becomes wider as the station terrain height decreases (Fig.12). The distance between the gray box of observed 10-meter wind speed and the pink box of the 10-meter wind speed predicted by WRF is greater as the station terrain height decreases. It shows that 10-meter wind speed predicted by WRF has less accuracy as the station terrain height decreases.*"

24. Lines 493-498: The use of the word "unstable" or "instability" in this section is confusing. I might say something more like "variability".
**Response:** Many thanks for your suggestion.
For clarity, we have corrected Lines 493-498 to the following text: "*With 1 km as the center, the measured 10-meter wind speed is more variable in areas where the station terrain height increases or decreases. However, the pink box of the 10-meter wind speed predicted by WRF being wider with the station terrain height decreases (Fig.12). The distance between the gray box and the pink box is longer with the station terrain height decreases. It shows that 10-meter wind speed predicted by WRF has less accuracy with the station terrain height decreases. The VMD-PCA-RF and VMD-PCA-lightGBM models significantly reduce the variability of the 10-meter wind speed predicted by WRF. When the height of the station increases or decreases at 1 km, the correction intensity tends to increase gradually. This further explains the higher importance of the height factor in the RF model training.*"

25. Fig. 14: The text claims this figure shows the actual wind speed in each month, but I cannot find that.
**Response:** Thank you very much for pointing this out.

For clarity, we have corrected the title of Fig. 14 to the following text: "*Figure 14: Evaluation histograms of 10-meter wind speed predicted by 10 models in different months in Experiment 1 and Experiment 2 ((a), (b), (c), (d), (e), and (f) represent R, FA (%), MAE (m/s), RMSE (m/s), rMAE (%), and rRMSE (%) respectively).*"

26. Line 513: Indexes > indices

**Response:** Many thanks for your suggestion. We've corrected Line 513 to the following text: "*As shown in the figure 14, the evaluation indices of the model trained in Experiment 2*"

---

## Author Comment (AC4)

**Response to reviewer comments**

Dear Editor and Reviewer,

We are very grateful for your time and valuable comments, which we found very helpful. We have addressed questions and comments raised by the reviewer in the revised manuscript with tracked changes. Please find our point-by-point response (in blue font) to the comments below. We hope our revisions have properly addressed your concerns.

Thanks again for your time.

Sincerely, The authors

**Reviewer 4**

General comments

The manuscript "A robust error correction method for numerical weather prediction wind speed based on Bayesian optimization, Variational Mode Decomposition, Principal Component Analysis, and Random Forest: VMD-PCA-RF (version 1.0.0)" by Zhou et al. introduces a hybrid method for correcting 10-meter wind speed predicted by WRF. The authors compare the performance of two sets of experiments with different predictors and report the best model for wind speed correction during December 2021 to January 2022. In general, this manuscript fits the scope of the Geoscientific Model Development. However, after reading the manuscript, I find it still has a few major flaws. Firstly, the descriptions for the observation data and methods are unclear and ambiguous, and some citations should be implemented in the main text. Secondly, the information in the main text, figures, and tables is repeated. For example, the authors just simply report many statistics for model validation and comparison in Section 3, which are also showed in the tables. I would suggest the authors to summarize the key points and analyze the potential reasons for the differences in the main text rather than listing the statistics, which can be better for readers' understanding. Finally, the writing and figures should be improved. Some figures should be combined, e.g., Figure 1 and 2. The captions for some figures are very simple, e.g., Figure 5, Figure 6, and Figure 10. The labels and legends might be enlarged for a better readability. This reviewer requests major revisions listed below. Response: Thank you very much for your recognition and encouragement of our work, we will further modify it according to your comments.

**Specific comments**

1. P5, Section 2.1: The description of the observation data is unclear. I would suggest the authors to give more details on this dataset. What are the data sources for the ground and satellite data? How do the authors process the data? How do the authors

interpolate the data across 410 sites? Please cite the data sources and related techniques.

**Response:** Thank you very much for pointing this out.

[revised manuscript text omitted]

3. P6, Line 162-166: Please add the citations for these WRF parameterizations and schemes.

**Response:** Thank you very much for pointing this out. We have deleted Line 162-166 in section 2.2.1, and we have added the citations in Table 1.

| Model (Version)     | WRF (V4.2)                                  |         |
|---------------------|---------------------------------------------|---------|
| Domains             | D1                                          | D2      |
| Horizontal grid     | 600*500                                     | 967*535 |
| points              |                                             |         |
| $\Delta x (km)$     | 9                                           | 3       |
| Vertical layers     | 58                                          |         |
| Longwave radiation  | RRTMG (Iacono et al., 2008)                 |         |
| Shortwave radiation | RRTMG (Iacono et al., 2008)                 |         |
| Land surface        | Noah LSM (Chen et al., 1997)                |         |
| Surface layer       | MYJ (Janjić, 1994)                          |         |
| Microphysics        | Thompson (Thompson et al., 2008)            |         |
| Boundary layer      | MYJ (Janjić, 1994)                          |         |
| Cumulus             | Tiedtke (Tiedtke, 1989; Zhang et al., 2011) |         |

Table 1: WRF configuration scheme

Chen, F., Janjić, Z., and Mitchell, K.: Impact of Atmospheric Surface-layer

Parameterizations in the new Land-surface Scheme of the NCEP Mesoscale Eta Model, Boundary-Layer Meteorology, 85, 391 – 421, https://doi.org/10.1023/A:1000531001463, 1997.

Iacono, M. J., Delamere, J. S., Mlawer, E. J., Shephard, M. W., Clough, S. A., and Collins, W. D.: Radiative forcing by long-lived greenhouse gases: Calculations with the AER radiative transfer models, J. Geophys. Res., 113, D13103, https://doi.org/10.1029/2008JD009944, 2008.

Janjić, Z. I.: The Step-Mountain Eta Coordinate Model: Further Developments of the Convection, Viscous Sublayer, and Turbulence Closure Schemes, Monthly Weather Review, 122, 927 – 945, https://doi.org/10.1175/1520-0493(1994)122<0927:TSMECM>2.0.CO;2, 1994.

Thompson, G., Field, P. R., Rasmussen, R. M., and Hall, W. D.: Explicit Forecasts of Winter Precipitation Using an Improved Bulk Microphysics Scheme. Part II: Implementation of a New Snow Parameterization, Monthly Weather Review, 136, 5095–5115, https://doi.org/10.1175/2008MWR2387.1, 2008.

Tiedtke, M.: A Comprehensive Mass Flux Scheme for Cumulus Parameterization in Large-Scale Models, Monthly Weather Review, 117, 1779–1800, https://doi.org/10.1175/1520-0493(1989)117<1779:ACMFSF>2.0.CO;2, 1989.

Zhang, C., Wang, Y., and Hamilton, K.: Improved Representation of Boundary Layer Clouds over the Southeast Pacific in ARW-WRF Using a Modified Tiedtke Cumulus Parameterization Scheme\*, Monthly Weather Review, 139, 3489–3513, https://doi.org/10.1175/MWR-D-10-05091.1, 2011.

4. P5 and P7: Figure 1 and Figure 2 both show the terrain heights in the study region. What's the difference between the two figures? I would suggest the authors to combine the two figures.

**Response:** Many thanks for your suggestion. We have combined the two figures. The revised figure is shown in Fig. 1 below.

---

## Referee Report (RR1)

In the abstract, it still says "Global Prediction System (GFS)" which is not correct.  The acronym is Global Forecast System.

Figure 1: It's pretty hard to see the provincial boundaries.  Partly because of the black dots, and partly because of the dark blue background.  Maybe they could be made thicker to better highlight the regions.

---

## Author Response (AR2)

**Response to reviewer comments**

Dear Editor and Reviewers,

We are very grateful for your time and valuable comments, which we found very helpful. We have addressed questions and comments raised by the reviewer in the revised manuscript with tracked changes. Please find our point-by-point response (in blue font) to the comments below. We hope our revisions have properly addressed your concerns.

Thanks again for your time.

Sincerely, The authors

Reviewer 2

The revised version of Zhou et al. addresses many of the reviewer concerns and is better presented than the original version. However, I still have major concerns about the manuscript as presented.

**Response:** Thank you very much for your recognition and encouragement of our work, we have taken your comments into consideration and further edited our manuscript.

General comments

1. The first major concern is of the limited timespan of the data sets used in training and testing. I understand that the authors explained that the dataset contains many hours of data over 410 weather stations, but finer sampling of the same period cannot fully compensate for the fact that the data is constrained to one particular season (December - February) of a particular year. I would suggest to at least conduct the testing (similar to what's done in Figure 13) in other seasons to extend to another winter period.

**Response:** Thank you very much for your advice. We have extended the testing to winter of 2022, and the evaluation results of the testing are shown in Fig. 13.

We have also added the following text on line 254: "While similar past studies for wind speed correction from NWP models usually use several years for training and at least one year for testing whereas our periods are shorter, the size of our data set is sufficient. For example, Sun et al. 2019 used a data set that contained 1827 days, from January 2012 to December 2016, using 143 grid points with a resolution of 0.5°\*0.5° predicted by ECMWF, followed by 24 features for each sample, with a training set size of 1827\*143\*24 for each prediction time. Meanwhile, the size of our training set is about 2160\*410\*12. Therefore, even though it only took us a month to train, for this project, we trained millions of data; Second, the training data we used here was obtained through daily operational runs of numerical weather forecasting,

---

## Author Response (AR3)

**Response to reviewer comments**

Dear Editor and Reviewers,

We are very grateful for your time and valuable comments, which we found very helpful. We have addressed questions and comments raised by the reviewer in the revised manuscript with tracked changes. Please find our point-by-point response (in blue font) to the comments below. We hope our revisions have properly addressed your concerns.

Thanks again for your time.

Sincerely, The authors

Reviewer 1

The revised version of the Zhou et al. manuscript addresses my concerns and I recommend it for publication.

**Response:** Thank you very much for your recognition and encouragement of our work, we have taken your comments into consideration and further edited our manuscript.

Specific comments:

1. My only suggestion is that instead of noting Time as UTC+08:00 in the figure legends, the figure axes "Hour" could be changed to "Local time (UTC+8)" or something similar which makes the figures more readable.

**Response:** Thank you very much for pointing this out.

We have changed the "Hour" of the figure axes to "Local time (UTC+8)" in the Figs. 6, 7, and 9.

The same changes have been corrected in the supplementary material.